# Identification of a Hydroxygallic Acid Derivative, Zingibroside R1 and a Sterol Lipid as Potential Active Ingredients of *Cuscuta chinensis* Extract That Has Neuroprotective and Antioxidant Effects in Aged *Caenorhabditis elegans*

**DOI:** 10.3390/nu14194199

**Published:** 2022-10-09

**Authors:** Shimaa M. A. Sayed, Saleh Alseekh, Karsten Siems, Alisdair R. Fernie, Walter Luyten, Christian Schmitz-Linneweber, Nadine Saul

**Affiliations:** 1Molecular Genetics Group, Institute of Biology, Faculty of Life Sciences, Humboldt University of Berlin, 10115 Berlin, Germany; 2Botany and Microbiology Department, Faculty of Science, New Valley University, El-Kharga 72511, Egypt; 3Max Planck Institute of Molecular Plant Physiology, 14476 Potsdam, Germany; 4Center for Plant Systems Biology and Biotechnology, 4000 Plovdiv, Bulgaria; 5AnalytiCon Discovery GmbH, 14473 Potsdam, Germany; 6Biology Department, Katholieke Universiteit Leuven, 3000 Leuven, Belgium

**Keywords:** *C. elegans*, *Cuscuta chinensis*, *Eucommia ulmoides*, healthspan, traditional Chinese medicine, cognitive fitness, ROS

## Abstract

We examined the effects of the extracts from two traditional Chinese medicine plants, *Cuscuta chinensis* and *Eucommia ulmoides,* on the healthspan of the model organism *Caenorhabditis elegans*. *C. chinensis* increased the short-term memory and the mechanosensory response of aged *C. elegans.* Furthermore, both extracts improved the resistance towards oxidative stress, and decreased the intracellular level of reactive oxygen species. Chemical analyses of the extracts revealed the presence of several bioactive compounds such as chlorogenic acid, cinnamic acid, and quercetin. A fraction from the *C. chinensis* extract enriched in zingibroside R1 improved the lifespan, the survival after heat stress, and the locomotion in a manner similar to the full *C. chinensis* extract. Thus, zingibroside R1 could be (partly) responsible for the observed health benefits of *C. chinensis*. Furthermore, a hydroxygallic acid derivative and the sterol lipid 4-alpha-formyl-stigmasta-7,24(241)-dien-3-beta-ol are abundantly present in the *C. chinensis* extract and its most bioactive fraction, but hardly in *E. ulmoides*, making them good candidates to explain the overall healthspan benefits of *C. chinensis* compared to the specific positive effects on stress resistance by *E. ulmoides*. Our findings highlight the overall anti-aging effects of *C. chinensis* in *C. elegans* and provide first hints about the components responsible for these effects.

## 1. Introduction

Human life expectancy has increased steadily over the past decades. Unfortunately, age-related diseases, such as neurodegenerative diseases and cognitive dysfunctions, increased in parallel [1,2,3,4]. In 2015, people over 60 years of age represented around 12% of the world’s population, and this percentage is expected to increase to 22% by the year 2050 [5,6]. Without a parallel increase in healthspan, the medical needs of the increasingly elderly population will have enormous economic and social consequences [7,8]. Age-related cognitive decline is one of the most severe health threats and risks affecting almost all elderly people around the world. In the United States, about 5.8 million Americans aged 65 and older lived with Alzheimer’s dementia in 2020. Several recent studies reported that the incidence of Alzheimer’s and other dementias in the United States and other Western countries may have declined in the past 25 years because of improved education quality and progress in prevention and control of risk factors [9,10]; however, the total number of people with Alzheimer’s or other dementias in those countries is expected to increase dramatically due to the growing number of people in the oldest age groups [11].

To prevent age-related diseases and dysfunctions efficiently, the aging process itself needs to be targeted. Many theories have been developed to explain the aging process, such as the oxidative stress theory of aging, which states that the accumulation of reactive oxygen species (ROS) over time leads to increased cell damage, physiological decline, and the appearance of age-related diseases [12]. The oxidative stress in cells and tissues is attributed to an overproduction of ROS, or an insufficient antioxidant defense, leading to oxidative damage [13,14]. Therefore, the search for effective antioxidant compounds to decelerate the aging process is an important strategy to prevent cognitive decline.

Traditional Chinese Medicine (TCM) offers several antioxidant and anti-aging preparations with a long medical history that is not restricted to Asian countries [15,16]. Unfortunately, the effectiveness and safety of these medical treatments are still not sufficiently established, not least due to their complex compositions and the large variety of bioactive components. Thus, quantitative and qualitative analyzes of these components is of utmost importance [17,18]. Seeds of *Cuscuta chinensis* Lam. (*C*. *chinensis*) are commonly used in TCM recipes for anti-aging, anti-inflammatory, anticancer, anti-apoptosis, anti-osteoporotic, and immunostimulatory effects, and as a liver and kidney tonic; they were recently shown to increase the healthspan of the model organism *Caenorhabditis elegans* (*C*. *elegans*) [16]. Especially because *C*. *chinensis* is a parasitic plant and its phytochemical composition is influenced by the host plant [19], the pharmacological and clinical effects of *C*. *chinensis* are subject to fluctuations and thus, are not easily determined. Several chemical studies have focused on the qualitative analysis of the major constituents in *C*. *chinensis* crude extracts by using different analytical methods such as HPLC and LC-MS/MS [17,18]. *C*. *chinensis* is rich in bioactive constituents, such as flavonoids, phenolic compounds, poly-saccharides, alkaloids, steroids, volatile oils, and lignans [20].

This study is a continuation of the work presented in Sayed, Siems, Schmitz-Linneweber, Luyten and Saul [16]. In addition to the previously shown general health benefits of the *C*. *chinensis* seed extract in the model organism *C*. *elegans*, we now focus on the antioxidant capacities, and the effects on sensory and cognitive abilities in this nematode. The *E*. *ulmoides* bark extract, which was shown to specifically improve the stress resistance in *C*. *elegans* [16], was used for comparison purposes. We hypothesize that the different anti-aging properties of *C*. *chinensis* and *E*. *ulmoides* are reflected in their chemical compositions. Thus, a UPLC-MS/MS analysis of the extracts was performed to identify single ingredients. Finally, three fractions enriched in ingredients from *C*. *chinensis* (astragalin, pinoresinol, and zingibroside R1) were tested for their ability to change the healthspan of *C*. *elegans*. Indeed, we found that the effects of zingibroside on health could at least in part explain the positive properties of the *C*. *chinensis* extract.

## 2. Materials and Methods

### 2.1. TCM Extracts Preparation, Fractionation and Purification of Single Compounds

The two TCM plants *Cuscuta chinensis* and *Eucommia ulmoides* were purchased from the Beijing Tong Ren Tang Chinese Medicine Company (Beijing, China). The organic extracts of *C*. *chinensis* and *E*. *ulmoides* were prepared by AnalytiCon Discovery GmbH (Potsdam, Germany) and stored under the batch numbers V-22579-W-00 and V-22582-W-00, respectively. The extraction process was carried out using a mixture of methyl ^t^butyl-ether (MTBE) and methanol (50:50) as well as with 100% methanol. Then the extracts were combined, dried and kept in the dark at 4 °C. Before further processing, all samples were completely dried and freshly dissolved. Thereafter, they were diluted with assay-buffer.

For the fractionation, 3.2 g from each extract was fractionated by using Reversed Phase High Performance Liquid Chromatography (RP-HPLC) with a LiChrospher-Select B column (Merck, Darmstadt, Germany; particle size:10 µm, diameter: 250 × 50 mm, flow rate 80 mL/min; solvent A: H_2_O, solvent B: methanol: acetonitrile (1:1); gradient from 23% B to 63% B in 57.7 min). The pH value was adjusted to 3 with formic acid. Every 30 s a new fraction was collected. For the purification of the phytochemicals astragalin, pinoresinol and zingibroside R1, the extract of *C*. *chinensis* was separated by MPLC and further fractionated by HPLC using reversed-phase modified silica (LiChrospher-Select B column, solvent A: ammonium formate—formic acid pH 3.0, solvent B: methanol-acetonitrile 50:50, gradient from 50% to 75% B in 60 min). Major compounds in the used fractions for this study were elucidated as astragalin, pinoresinol, and zingibroside R1, respectively, based on LCMS and H-NMR by comparison with analytical data of pure compounds. The purity of the compounds could be estimated from H-NMR spectra: Astragalin approx. 75%, Pinoresinol approx. 75%, and Zingibroside approx. 50%. The structures were elucidated by mass spectrometry. The fractions enriched in astragalin, pinoresinol, and zingibroside R1 were deposited and stored at AnalytiCon Discovery GmbH under the codes C-3071-I-A06, C-3071-I-A10 and C-3071-L-D03, respectively. For LCMS, the HPLC Shimadzu LC-30AD prominence (Shimadzu Deutschland GmbH, Duisburg, Germany) with PDA and light scattering detection (Sedex 85) and MS Shimadzu 2020 Single quadrupole (Shimadzu Deutschland GmbH, Duisburg, Germany) was used. HPLC-conditions: column lunaC8(2) 3 µm, 50 × 2 mm, solvent: A: 5 mM ammonium formate + 0,1% formic acid, solvent B: methanol: acetonitrile (1:1) + 5 mM ammonium formate + 0.1% formic acid, gradient: from 5% B to 100% B in 4 min, 2 min 100% B hold, flow rate 0.7 mL/min, injection volume 5 µL, detection ELSD (Sedex85, pressure 4 bar, nebulizer temp. 35 °C), scan area (MS): 100–1400 amu, pos/neg switch, PDA 200–400 nm) and NMR spectroscopy (BRUKER Avance 400 MHz, Topspin 4.0, (BRUKER Biospin AG, Ettlingen, Germany) solvent methanol-d4. For acquisition parameters, see NMR spectra in S7, S9, and S11).

### 2.2. UPLC-MS/MS Analysis of C. chinensis and E. ulmoides Extracts and Fractions

The plant extracts and fractions were dissolved in dimethyl sulfoxide (DMSO) at a stock concentration of 60 mg/mL. Fifty μL of these plant extract solutions were added to 1 mL of an LC-grade water-methanol mixture (1:1). After sonication of the samples for 10 min in an ice-cooled sonicator bath, tubes were centrifuged for 15 min. Thereafter, 150 μL of the supernatant was transferred to LC tubes for analysis. The samples were run on a UPLC-MS instrument as described previously [21] with a few modifications. The Acquity UPLC system (Waters GmbH, Eschborn, Germany) was equipped with an HSS T3 C18 reversed-phase column (100 × 2.1 mm internal diameter, 1.8 μm particle size; Waters GmbH, Eschborn, Germany) that was operated at a temperature of 40 °C. The mobile phases consisted of 0.1% formic acid in water (solvent A) and 0.1% formic acid in acetonitrile (solvent B). The flow rate of the mobile phase was 400 μL/min, and 2 μL of sample was loaded per injection. The UPLC instrument was connected to an Exactive Orbitrap-focus (Thermo Fisher Scientific, Waltham, MA, USA) via a heated electrospray source (Thermo Fisher Scientific, Waltham, MA, USA). The spectra were recorded using full-scan positive and negative ion-detection mode, covering a mass range from *m*/*z* 100 to 1500. The resolution was set to 70,000, and the maximum scan time was set to 250 ms. The sheath gas was set to a value of 60 while the auxiliary gas was set to 35. The transfer capillary temperature was set to 150 °C while the heater temperature was adjusted to 300 °C. The spray voltage was fixed at 3 kV, with a capillary voltage and a skimmer voltage of 25 V and 15 V, respectively. MS spectra were recorded from minutes 0 to 19 of the UPLC gradient. Processing of chromatograms, peak detection, and integration were performed using RefinerMS (version 5.3; GeneData, Basel, Switzerland) and Xcalibur software (version 4.0; Thermo Fisher Scientific, Waltham, MA, USA). Metabolite identification and annotation were performed using standard compounds, data-dependent method (ddMS2) fragmentation, literature and metabolomics databases [22].

### 2.3. Caenorhabditis elegans Maintenance

The wild-type *C*. *elegans* strain N2 (Bristol), the transgenic *C*. *elegans* strain JV1 (jrIs1 [rpl-17p::HyPer + unc-119(+)]) as well as the *Escherichia coli* feeding strain OP50 were obtained from the *Caenorhabditis* Genetics Center (CGC) (Minneapolis, MN, USA). Nematodes were maintained according to Brenner [23] and according to our previous study [16] at 22 °C on 96 mm nematode growth medium (NGM) agar plates seeded with live OP50 bacteria, which were grown at 37 °C and concentrated to OD_595_ = 5 beforehand. Synchronized and contamination-free populations were regularly generated by lysing young adults in a 3% sodium hypochlorite solution until eggs were isolated, based on a protocol from Stiernagle [24]. The obtained eggs hatched in M9 buffer overnight, and were transferred to new NGM plates the following day.

### 2.4. The Treatment of C. elegans

The plant extracts and fractions were dissolved in dimethyl sulfoxide (DMSO) at a stock concentration of 60 mg/mL and 30 mg/mL, respectively, and then added to the NGM agar plates as well as the OP50 bacteria at a final concentration of 30 µg/mL and 15 µg/mL, respectively. The selected extract concentration was inspired by different health- or lifespan promoting extracts, which were successfully used in the nematode in a range of 25–50 µg/mL, such as the Alaskan chaga and cranberry extract [25], *Anacardium occidentale* extract [26], and Korean mistletoe [27]. The fractions enriched in astragalin and pinoresinol were dissolved in DMSO and used at a final concentration of 2 µg/mL in NGM agar plates and OP50 bacteria. The molecular weight of Zingibroside R1 is about twice as high compared to astragalin and pinoresinol; thus, double the amount was used for the fraction enriched in zingibroside R1 (4 µg/mL). DMSO (0.05%) was used as a control in all experiments, and carbenicillin (2 mg/mL) was added to all agar plates. Synchronized, untreated L4 larvae were transferred to the prepared NGM agar plates, followed by the addition of 100 µM 5-fluorodeoxyuridine (FUdR), which prevents the development of progeny [28]. The nematodes were incubated on those plates at 22 °C until the respective adulthood stages, at which the following experiments were performed.

### 2.5. Chemotaxis Assay

The chemotaxis assay was carried out with *C*. *elegans* on the 3rd and 7th day of adulthood, as previously described by Margie et al. [29], with slight modifications. The chemotaxis assay plates, containing NaCl-free NGM, were divided into four quadrants, referred to as two control and two NaCl areas. Equally-sized NGM agar plugs spiked with 100 mM NaCl were placed on the NaCl-spots (N) as shown in Figure 1A, and incubated overnight. To start the assay, approximately 150 adult TCM-treated or control worms (washed twice with CTX buffer) were placed in the center of the plate, and left to move freely on the assay plates. One µL of 1 M sodium azide was applied on each spot in order to immobilize the worms once they reached the area. Thereafter, the assay plates were incubated for one hour at 22 °C. Then, worms in each quarter were counted, and the chemotaxis index (CI) was calculated with CI = (n [N-quadrants] − n [C-quadrants])/n [N-quadrants + C-quadrants], whereby nematodes that had not left the starting circle (S) were not included in the calculation. Chemotaxis assays were carried out in triplicate on separate chemotaxis plates.

### 2.6. Learning and Short-Term Associative Memory

The ability of *C*. *chinensis* and *E*. *ulmoides* to improve the associative learning and memory of *C*. *elegans* was tested according to Kauffman et al. [30]. The diagram presented in Figure 2A briefly shows the procedures of the positive associative olfactory learning and memory assays. Seven-day old adult worms were collected, starved in M9 buffer for one hour, and then exposed to conditioning massed training. Illustratively, they were transferred to NGM plates containing OP50 mixed with 10% butanone. The plates were incubated at 22 °C for 30 min and the worms were tested before (I) and after (trained) butanone conditioning for chemotactic abilities as described above, but instead of NaCl, butanone was used as attractant and added directly to the B-spot (1µL of 10% butanone). The learning index (LI) was calculated according to LI = CI_Trained_ − CI_Naive_.

For the estimation of the short-term associative memory (STAM), butanone-trained and naive worms were transferred to NGM plates with OP50 but without butanone (holding plates). After an incubation at 22 °C for specified intervals (30 min or 2 h), the worms were tested for chemotaxis to butanone, and the associative memory index (AMI) was calculated with AMI = CI_Trained_ − CI_Naive_.

### 2.7. Behavioral Responses of C. elegans to Mechanical Stimuli

The behavioral response assay of elderly worms (12th day of adulthood) was carried out blinded by using a sterilized hair to gently touch the worm’s body at the anterior and posterior ends according to Chalfie et al. [31]. The anterior and posterior ends were touched alternately (five times per end), and the positive responses (moving backwards or retracting the head after touching the anterior end, or moving forward or stretching out the head after touching the posterior end) were tabulated. The response rate was determined for 20 worms per treatment.

### 2.8. Survival Assays under Stress Conditions

The oxidative stress assay was carried out according to Peixoto et al. [32]. On the 12th day of adulthood, approximately 75–90 nematodes per treatment group were distributed on three NGM plates (excluding the test-extracts) containing 60 mM paraquat, a known inducer of oxidative stress [33,34,35] and incubated at 22 °C (oxidative stress assay). The heat stress assay was also performed with nematodes on the 12th day of adulthood. The worms were exposed to heat stress (37 °C) for 3 h with subsequent incubation at 22 °C on the treatment plates. The number of surviving and dead worms was monitored daily until all had died. The worms were considered dead when they failed to respond to a gentle touch. These assays were performed blinded two or three times.

### 2.9. Reactive Oxygen Species (ROS) Measurements

The strain JV1, which expresses the YFP-based hydrogen peroxide sensor “HyPer” was used to ascertain the endogenous ROS level according to Back et al. [36]. On the 12th day of adulthood, about 25 individuals per treatment group were transferred to a 2% agarose pad on a microscope glass slide, and immobilized using 1 M sodium azide. The Axiolab fluorescence microscope (Carl Zeiss, Jena, Germany) equipped with a YFP filter set, a ProgRes C12 digital camera (Jenoptik, Jena, Germany), and an objective with 10× magnification was used to image the worms. Mean fluorescence intensities per single worm were quantified using the CellProfiler software [37]. The intensity values were normalized by subtracting the yellow autofluorescence values measured in extract-treated and control worms of the same age. DMSO-treated worms exposed to 10 mM H_2_O_2_ for 30 min on day 12 of adulthood were used as a positive control. Three independent experiments were performed per treatment group.

### 2.10. Reproduction Assay

The effect of the extracts and selected fractions on fecundity was determined by counting the total offspring per nematode. Ten synchronized L4 worms, which were treated with the herbal preparations since L1 larval stage, were transferred individually to treatment and control plates at 22 °C seeded with OP50 bacteria but without the reproductive inhibitor FudR. Every 24 h, each single worm was moved to a new plate until the 3rd day of adulthood. The hatched nematodes were counted after they developed to L2 or L3 larvae.

### 2.11. Lifespan Assay

For each treatment group, about 80–100 synchronized L4 larvae were transferred to three small NGM agar plates seeded with OP50 and containing the test-compound. The plates were incubated at 22 °C and the nematodes were transferred to fresh treatment plates every seven days. Surviving and dead worms were counted daily until all worms had died. Ruptured animals as well as nematodes which left the agar surface were censored.

### 2.12. Swimming Behavior

On the 12th day of adulthood, five worms per group were transferred to wells with a depth of 0.5 mm and a diameter of 10 mm on a microscope slide, which were filled with M9 buffer and covered by a cover slip to facilitate visualization. Then, one 60-s video with a magnification of 10× was recorded per well, and ≥50 nematodes were monitored per treatment and age. After isolating every second frame of the videos, and applying the greyscale and invert mode via Adobe Photoshop (version 19.1.7; Adobe Inc., San José, CA, USA), the wave initiation rate, activity index, brush stroke and body wave number were determined with the CeleST software [38].

### 2.13. Statistical Analysis

The results were statistically analyzed using a one-way ANOVA test, followed by the Bonferroni’s multiple comparison test available online (https://astatsa.com/OneWay_Anova_with_TukeyHSD/; accessed on 12 March 2022). Survival and lifespan assays were statistically analyzed using a log-rank test via the Online Application for Survival analysis OASIS 2 [39] with subsequent Bonferroni correction. Data are displayed as mean ± SEM (standard error of the mean). Differences were considered statistically significant if their *p*-value was * (*p* < 0.05), ** (*p* < 0.01), *** (*p* < 0.001) or **** (*p* < 0.0001).

## 3. Results

### 3.1. C. chinensis and E. ulmoides Did Not Modify Chemotactic Abilities

Sensory perception after the treatment with *C*. *chinensis* and *E*. *ulmoides* extract was evaluated by a salt chemotaxis assay in *C*. *elegans* on the 3rd and 7th day of adulthood. Due to severe movement restrictions, older individuals were not studied with this assay. *C*. *elegans* links the environmental salt concentration during its cultivation to the presence of food. Thus, the worms navigate along the salt gradient on the prepared chemotaxis plates to find the desired food on the NaCl-spots (Figure 1A); this ability significantly fades with increasing age in all treatment groups (Figure 1B). The chemotaxis index for worms treated with *C*. *chinensis* was 0.72 and 0.55 on the 3rd and 7th days of adulthood, respectively, and for the *E*. *ulmoides*-treated group 0.72 and 0.63 on the 3rd and 7th days of adulthood, respectively. The control group showed similar chemotactic capacities with a CI of 0.74 and 0.60, respectively (Figure 1B). Thus, neither of the two extract treatments led to a modification of the chemo-attractive response compared to the untreated worms.

### 3.2. C. chinensis Improved the Short-Term Associative Memory

We examined the ability of *C*. *chinensis* and *E*. *ulmoides* to change the cognitive fitness of *C*. *elegans* on the 7th day of adulthood by measuring associative learning and short-term associative memory (STAM). Temporarily starved extract-treated and untreated worms were fed with OP50 in the presence of butanone for 30 min (Figure 2A). Thus, the nematodes learn that butanone, which normally elicits a low chemotactic response [40], is associated with food and they should therefore be attracted to it. The associative learning index (LI) of the wild type worms treated with *C*. *chinensis* and *E*. *ulmoides* did not show statistically significant differences compared to the untreated nematodes. The LI of *C*. *elegans* treated with *C*. *chinensis* and *E*. *ulmoides* had a value of 2.11 and 2.32, respectively, compared to 2.09 for the control worms (Figure 2B). These slight increases were, however, not significant.

Interestingly, the application of *C*. *chinensis* significantly increased the STAM index (Figure 2C). After the butanone training period, the nematodes were placed for 30 and 120 min on holding plates, which contained OP50 without butanone (Figure 2A). *C*. *chinensis* pre-treated nematodes could remember the previously trained butanone attraction better after both holding periods, with an increase of 50 and 78% compared to the control (Figure 2C). By contrast, the treatment with *E*. *ulmoides* did not lead to significant changes of the STAM index (Figure 2C). Thus, the short-term associative memory is positively modulated specifically by the *C*. *chinensis* extract.

### 3.3. Cuscuta chinensis Increased the Mechanosensory Response of C. elegans

We studied the ability of the TCM extracts to change the *C*. *elegans* mechanical sensory response to gentle touches on the 12th day of adulthood. In these elderly individuals, mechanosensory responses are already impaired [25]. This impairment was attenuated by treatment with *C*. *chinensis*, which showed a behavioral response rate to anterior and posterior gentle touches of 97 and 84%, respectively, whereas the control only featured a rate of 83 and 60%, respectively (Figure 3). However, the treatment with *E*. *ulmoides* did not exhibit any significant effect on the mechanosensory response, with a response rate of 85% for anterior and 73% for posterior touches (Figure 3). In conclusion, the *C*. *chinensis* extract enables a better mechanosensory response in aging *C*. *elegans*.

### 3.4. C. chinensis and E. ulmoides Increased the Oxidative Stress Resistance

The ability of *C*. *chinensis* and *E*. *ulmoides* to enhance the survival of *C*. *elegans* on the 12th day of adulthood under oxidative stress is shown in Figure 4. Worms fed with *C*. *chinensis* and *E*. *ulmoides* since their L4 larval stage exhibited significant improvements in the resistance against 60 mM paraquat (inducer of oxidative stress). The treatment with *C*. *chinensis* and *E*. *ulmoides* could significantly prolong the mean survival of *C*. *elegans* to 2.85 and 2.86 days, respectively, compared to the control, which survived only 2.41 days (Figure 4). Moreover, Table 1 illustrates that there is an increase in the minimum survival in the *C*. *chinensis* and *E*. *ulmoides* treatment groups (to 1.17 and 1.27 days, respectively compared to 0.84 days in the control group). The increased survival after paraquat treatments is indicative of a better oxidative stress resistance in older worms.

### 3.5. TCM Extracts Decreased the Endogenous ROS Level

To determine the endogenous oxidative stress level, we used a transgenic strain that expresses the biosensor HyPer (hydrogen peroxide-specific sensor), which selectively detects H_2_O_2_. Interestingly, the level of intracellular H_2_O_2_ was significantly decreased in the *C*. *chinensis* and *E ulmoides* treatment groups on the 12th day of adulthood by 67% and 34%, respectively, compared to the control (Figure 5). Nematodes treated with 10 mM H_2_O_2_ (positive control), which is a known inducer of oxidative stress, exhibited the highest fluorescence intensities (Figure 5). Thus, both extracts are able to decrease the oxidative stress level in aged nematodes.

In addition, the popular H_2_DCFDA assay was used to analyze the endogenous ROS level. Interestingly, the extract treatments increased the ROS level in this assay. In line with Labuschagne et al. [13], the results were, however, deemed rather unreliable as shown and explained in Appendix A.

### 3.6. Only One C. chinensis Fraction Improved Locomotion, Mechanosensation as well as Oxidative Stress Resistance

Due to the wealth of possible originators of the observed health benefits, bioassays with extract fractions were performed in addition to the full extract. We hypothesized that there should be at least one fraction per extract which triggers the health benefits to a greater extent compared to the other fractions. The analysis of the chemical composition of these most active fractions would take us a step closer to the identification of the responsible single components.

Therefore, each extract was fractionated using RP-HPLC. 20 fractions of the *E*. *ulmoides* extract were tested for their ability to change the survival of *C*. *elegans* after heat stress (37 °C for 3 h) on the 12th day of adulthood. The heat stress assay was selected due to the better survival of *E*. *ulmoides*-treated worms after heat shock [16]. Among the twenty fractions, seven were able to improve the survival in nematodes to a similar extent as the full extract. Compared to the control, worms treated with *E*. *ulmoides* fractions no.1, 5, 7, 8, 9, 17, and 20 showed an increased survival by 21, 27, 19, 18, 24, 21, and 17%, respectively (see Appendix A, Appendix A), whereas the full extract showed an increase of 25%. Furthermore, treatment with fractions 6 and 16 led to slight but significant survival benefits by 9 and 10%.

Next, 17 fractions of the *C*. *chinensis* extract were tested for their ability to change the physical fitness in *C*. *elegans* on the 12th day of adulthood. *C*. *chinensis*-treated nematodes showed several health and fitness improvements, with the enhancement of the swimming behavior being one of the strongest effects [16]. Thus, the swim assay was selected to test the ability of the fractions to change locomotor fitness by measuring wave initiation rate, activity index, brush stroke, and body wave number. The wave initiation rate is the number of body waves per minute, which indicates the movement-speed, whereas the body wave number quantifies the waviness of the body at each time point. The activity index adds up the number of pixels that are covered by the nematode during the time spent for two strokes, as an indicator for the vigorousness of bending. Finally, the brush stroke parameter reflects the area covered by the body in a complete stroke, which indicates the depth of the movement. These features were selected due to their strong age dependence, since the body wave number increases, and the remaining parameters decrease with age [38]. Treatment with the *C*. *chinensis* extract led to an improvement of all these parameters (see Appendix A, Appendix A). Interestingly, only fractions B, E, F, K, N and O led to significant enhancement of the activity index. However, almost all tested fractions improved the wave initiation rate, brush stroke, and body wave number, whereas fractions B, E, F, I, K, N, and O triggered the strongest changes.

According to these results, seven fractions of *E*. *ulmoides* (no.1, 5, 7, 8, 9, 17 and 20) and of *C*. *chinensis* (B, E, F, I, K, N and O) were selected for testing in greater detail.

First, the impact of the selected fractions on the *C*. *elegans* neuronal fitness on the 12th day of adulthood was examined. *C*. *chinensis* fractions no. I, N and O as well as the full extract exhibited the best impact on improving the mechanical sensory response in *C*. *elegans*, compared to DMSO-treated worms. On the other hand, none of the selected *E*. *ulmoides* fractions elicited a significant change of the mechanosensory response (see Appendix A, Appendix A).

Furthermore, to determine the stress resistance of fraction-treated nematodes, an oxidative stress test was performed. Interestingly, only worms fed with *E*. *ulmoides* fraction 5 and with *C*. *chinensis* fraction O, as well as with the full extracts, showed an improved survival during paraquat exposure. *C*. *chinensis* extract and its fraction O increased the mean survival from 2.3 for untreated worms to 2.89 and 3.06 days, respectively (see Appendix A, Appendix A). Moreover, *E*. *ulmoides* and its fraction 5 significantly prolonged the mean survival to 2.94 and 2.87 days, respectively.

Finally, to uncover potential negative side effects of substance exposure, a reproduction assay was conducted by counting the total offspring of the treated and untreated animals. None of the fractions and extracts significantly decreased the reproductive output, although *C*. *chinensis* extract and its fractions N and O showed a slight trend of reduced offspring. Surprisingly, four *E*. *ulmoides* fractions (7, 8, 9, 17) and the *C*. *chinensis* fraction K significantly increased the total offspring by 9–13% (see Appendix A, Appendix A).

In summary, the fractionation of plant extracts helped attribute certain observed health improvements to selected fractions of the total extracts, an important step towards the identification of individual active compounds. Only the *C*. *chinensis* fraction O was able to enhance all parameters, i.e., locomotion, mechanosensation and the survival during oxidative stress, in aged nematodes. None of the *E*. *ulmoides* fractions could trigger an improved mechanosensory response. However, fraction no. 5 was most effective for improving stress resistance.

### 3.7. A Hydroxygallic Acid Derivative Is the Most Abundant Component in C. chinensis Seeds

To identify promising single components as potential initiators of the observed health effects of the extracts, we separated the samples via ultraperformance liquid chromatography followed by tandem mass-spectrometry and the analysis of the resulting spectra via the software Xcalibur (Thermo Fisher Scientific). The metabolites were identified using standard compounds, data-dependent method (ddMS2) fragmentation, literature and metabolomics databases.

First, we wanted to identify the most abundant single components in the *C*. *chinensis* extract. We focused on *C*. *chinensis* due to its overall healthspan improvement shown in our previous paper [16] and current report. *E*. *ulmoides*, which only specifically enhanced the stress resistance in *C*. *elegans*, was included in our analysis for comparison purposes. Furthermore, *C*. *chinensis* fraction O was included as the most effective healthspan-promoting fraction with the broadest effects. That may mean that it contains many (if not most) of the bioactive compounds responsible for the different effects of the *C*. *chinensis* extract.

The most abundant compound In the *C*. *chinensis* extract was a hydroxygallic acid (2,3,4,5,-Tetrahydroxybenzoic acid) derivative representing more than 5% of the total peak area, while only about 0.67% of the peak-area in the *E*. *ulmoides* extract (Table 2). Unfortunately, the second most abundant component, which features a similar distribution in the *C*. *chinensis* extract, could not yet be identified.

Next to the most abundant compounds, we also were interested in the biggest differences between the two plant extracts. Here, 4-alpha-formyl-stigmasta-7,24(24^1^)-dien-3-beta-ol stands out, which constituted around 0.8% of the total peak area in the *C*. *chinensis* extract and which could not be detected in the *E*. *ulmoides* extract at all (Table 2). Furthermore, 2% and 1% of the peak area in the *C*. *chinensis* extract were identified as a dienoic acid derivative and coumarin derivative, respectively, which are only present in trace amounts (about 0.002% of the total peak area) in the *E*. *ulmoides* extract. Several other abundant (each about 1% of the peak area) compounds in the *C*. *chinensis* extract, such as a cinnamic acid derivative, hexadecenoic acid, d-pinoresinol-4-O-glucoside, and stearic acid (or hexadecenoic acid), exhibit nearly a tenfold lower relative presence in the *E*. *ulmoides* extract.

In a third step, we screened the spectra obtained by UPLC-MS/MS analysis for further known bioactive compounds [20]. Several bioactive compounds which were classified and identified as flavonoids (quercetin and its derivatives; kaempferol and its derivatives; astragalin; apigenin; isorhamnetin-*O*-glucoside; and rutin), phenolic acids (caffeoylquinic acid derivative and chlorogenic acid derivative), and lignans (lignan-coumaroylglucoside and secoisolariciresinol) are present in small amounts in both extracts (Table 3). Interestingly, the *E*. *ulmoides* extract seems to be relatively richer in these select bioactive compounds compared to *C*. *chinensis*. Especially, quercetin and its derivates, lignan-coumaroylglucosides, caffeoylquinic acid derivatives as well as isorhamnetin-3-O-glucoside are more abundant in *E*. *ulmoides*.

In our analysis of the *C*. *chinensis* fraction O, we sought to identify differences and commonalities with the full extract. Compounds not present in fraction O are likely not of high relevance for the health effects and can thus be eliminated from future analyses. Interestingly, *C*. *chinensis* fraction O features a relatively high percent peak area (about 4.7%) of the hydroxygallic acid derivative, similar to the top peak in the full extract. All known bioactive compounds described above were also found in *C*. *chinensis* fraction O, but only in trace amounts. In addition, several compounds which are highly abundant in the full extract were also found in fraction O, such as d-Pinoresinol-4-O-glucoside, a cinnamic acid derivative, hexadecenoic acid, and zingibroside R1. A number of highly abundant compounds in the full extract are, however, strongly reduced in fraction O. This includes 2-C-Methyl-D-erythritol 4-phosphate, the dienoic acid derivative, and coumarin derivative. Altogether, the comparison of the two full plant extracts and fraction O points towards a number of promising single compounds for healthy aging effects, i.e., a hydroxygallic acid derivative and 4-alpha-formyl-stigmasta-7,24(24^1^)-dien-3-beta-ol. The full set of LC-MS data are presented in Appendix A. 

### 3.8. Fractions Enriched in Astragalin and Zingibroside R1 Extend the Lifespan of C. elegans

To understand the importance of the single components in the health improvement of aged worms, they need to be analyzed separately in different health assays. Here, we began by focusing on three single compounds, namely astragalin, pinoresinol, and zingibroside R1. Fractions enriched in these compounds were obtained from the organic extract of *C*. *chinensis* seeds and the chemical structures of the identified compounds are shown in Figure 6. UPLC-MS and NMR spectra for the fractions are shown in Appendix A (Appendix A).

We first tested whether these three fractions have a general effect on life span, given that the *C*. *chinensis* treatment showed a significant prolongation of the mean lifespan by 10.4% (Table 4, Figure 7). When treated with the fractions enriched in astragalin and zingibroside R1, the mean lifespan of the worms increased by 8.1 and 10%, respectively. However, the pinoresinol-containing fraction exerted no significant effect on the lifespan of *C*. *elegans* (Table 4, Figure 7). Furthermore, the astragalin and zingibroside R1 fractions increased the maximum lifespan to 30 days, compared to the 28 days of the control group (Table 4). Thus, astragalin and zingibroside R1 could be involved in the life-prolonging ability of the full *C*. *chinensis* extract.

### 3.9. Fractions Enriched in Astragalin, Pinoresinol and Zingibroside R1 Increased Heat Stress Resistance of C. elegans

We next tested the potential of the *C*. *chinensis* extract and the fractions enriched in astragalin, pinoresinol and zingibroside R1 for their ability to improve the survival of *C*. *elegans* after heat stress exposures on the 12th day of adulthood. Treatment of the nematodes with astragalin, pinoresinol and zingibroside R1 significantly increased the mean survival by 19.2, 12.7, and 12.1%, respectively, compared to control (Table 5, Figure 8). Interestingly, *C*. *chinensis*, which was already known to boost the survival after heat stress [16], showed the most effective survival improvement by boosting the mean, minimum, medium, and maximum *C*. *elegans* survival after heat stress (Table 5).

### 3.10. Fraction Enriched in Zingibroside R1 Improved Swimming Behaviour

The effect of *C*. *chinensis* and its fractions enriched in astragalin, pinoresinol, and zingibroside R1 on the locomotor fitness of *C*. *elegans* was investigated on the 12th day of adulthood, as previously explained (Section 3.6). Treatment with the *C*. *chinensis* extract was previously shown to enhance four parameters of swimming behavior in aged *C*. *elegans* [16], which was replicated in this study (Figure 9). Interestingly, the fraction enriched in zingibroside R1 improved the movement capacities in all tested parameters compared to control: the activity index, wave initiation rate and brush stroke of the zingibroside R1-treated worms increased by 37%, 39%, and 29%, respectively (Figure 9). Furthermore, the fraction enriched in zingibroside R1 significantly decreased the body wave number by 19%, which is indicative of improved physical fitness. On the other hand, the fractions enriched in astragalin and pinoresinol did not elicit any significant effect on the swimming behaviour of *C*. *elegans* (Figure 9).

## 4. Discussion

The aging process is strongly linked to neurodegenerative diseases, such as Alzheimer’s and Parkinson’s, as well as to cognitive decline [41,42]. Many studies reported that lifestyle, environmental, and genetic factors could play an important role in preventing age-related neurodegenerative disorders, but no effective cure or prevention is available, thus far [7,42,43,44]. Medicinal plants as well as complementary and alternative medicine, which are already used widely in most countries [44,45,46,47,48], could be a valuable source of new drug candidates to improve cognitive function and to prevent neurodegenerative disorders [42,49,50,51]. One promising candidate is the *C*. *chinensis* extract and its bioactive compounds.

### 4.1. Neuroprotective Potential of C. Chinensis

Although *C*. *elegans* has only 302 neurons, which cannot display the complexity of a human brain with its 100 billion neurons, the small worm was proven to be a suitable model organism in the search for (natural) substances with neuroprotective characteristics. *C*. *elegans* shows associative and non-associative behavior, and has both short- and long-term memory [52,53,54]; thus, it is frequently used to pre-screen compounds for their later pharmaceutical use against neurodegenerative diseases in higher animals [55]. In this study, we discovered the neuroprotective potential of *C*. *chinensis*.

Deterioration in neuronal synaptic transmission is one of the age-related effects which occurs in old nematodes, and which is associated with weakened locomotion skills and mechanosensation abilities [56,57]. We observed that *C*. *chinensis* improved the mechanical sensory response in aged worms by significantly increasing the anterior and posterior touch response after a gentle touch. This observation may be linked to altered morphology of mechanosensory neurons during aging, as described by Scerbak et al. [25]. They studied the effect of different Alaskan berry species and Chaga mushrooms on the function of touch receptor neurons in *C*. *elegans* at various ages; bog blueberry and lowbush cranberry increased the anterior and posterior touch response on the 11th day of adulthood, while the fungal treatment significantly augmented the anterior touch response. Interestingly, the three treatments provoked different morphological changes in mechanosensory neurons.

In order to study the effect of the extract treatments on the touch response in more detail, imaging of touch receptor neurons could be helpful as shown in Toth et al. [58]. The morphological changes of the ALM and PLM neurons during aging and the potential protective effect of *C*. *chinensis* to maintain the structural integrity of the cells can be made visible via fluorescence labeling in several transgenic *C*. *elegans* strains. In addition, the imaging of synaptic vesicle markers in these neurons, such as RAB-3 [59], could be helpful to evaluate the influence of the extract treatment on synaptic transmission. Finally, the potential neuroprotective effects of strong antioxidants could be analyzed in addition, in order to evaluate whether the antioxidative properties of *C*. *chinensis* are responsible for the neuroprotective effects.

To examine the influence of the TCM extracts on learning and memory in *C*. *elegans*, we used the AWC neuron-sensed odorant butanone. No enhancement of the learning index was observed, which might be attributed to the short incubation period with butanone (only 30 min). At the tested age (7th day of adulthood), olfactory functions are already impaired in *C*. *elegans*, leading to anosmia, which is considered one of the earliest symptoms of neurodegeneration [30,60]. It is therefore conceivable that longer incubation periods or the use of stronger odorants could change the outcome of this assay.

Nevertheless, a beneficial effect of the *C*. *chinensis* extract was apparent in the memory assay. In *C*. *elegans* and other invertebrates, such as *Aplysia*, *Drosophila* and *Hermissenda*, two forms of memory are known, i.e., short-term memory that can range from seconds to hours, and long-term memory, which covers hours to days or even weeks [61]. Our study focused on short-term memory after a single (“massed”) training and after two different “holding” time periods; *C*. *elegans* treated with the *C*. *chinensis* extract better retained their “learned” association after both tested holding periods (30 min and 2 h) compared to control. This is in line with Lin et al. [62], who reported that *C*. *chinensis* extract protected against scopolamine-induced memory deficit in mice. Another study noted that *C*. *chinensis* is able to improve memory impairment of rats caused by cerebral ischemia [63,64], further consolidating the results of this assay.

Interestingly, *C*. *chinensis* did not improve chemotactic abilities compared to control worms. When *C*. *elegans* ages, the chemo-attraction to NaCl deteriorates; therefore, the chemotaxis index on the 7th day of adulthood was lower than on the 3rd day of adulthood. This deterioration was visible in all treatment groups. Several studies showed that it is possible to mitigate this decline with natural substances. To name a few, 1–1000 µg/mL *Ocimum sanctum* extract [65], 25 µM alpha-lipoic acid and 25 µM epigallocatechin gallate [66], 5 µM withanolide A [67], 1–100 µg/mL *Bacopa monnieri* extract [68], and 0.01 µg/mL Ayurvedic polyherbal extract [69] were all able to increase the chemotaxis index in *C*. *elegans* around the 5th day of adulthood. The extract concentration tested in our study (30 µg/mL) is comparable to those tested in the *O*. *sanctum* and *B*. *monnieri* study, and the age of nematodes during the assay is quite similar. These parameters are therefore unlikely to account for our negative result. Nevertheless, it is conceivable that *C*. *chinensis* could boost chemotactic abilities at higher concentrations or in a different age class. Thus, a dose response curve using different *C*. *chinensis* extract concentrations in a few selected health assays would be helpful to determine the optimal concentration.

To summarize, in addition to improved locomotion, lifespan, pharyngeal pumping, and stress resistance [16], *C*. *chinensis* is also capable of enhancing cognitive function by increasing short-term memory and mechanosensation. The failure of *E*. *ulmoides* to alter cognitive fitness is in line with our previous report, which suggested that *E*. *ulmoides* is a specific stress resistance enhancer.

### 4.2. Antioxidative Features of C. chinensis and E. ulmoides

The accumulation of ROS inside cells leads to oxidative stress and is linked to various age-related and neurodegenerative diseases [13]. The beneficial impact of *C*. *chinensis* and *E*. *ulmoides* on the resistance against thermal and pathogenic stress was already reported in our previous publication [16]. Here, we additionally evaluated the ability of these plant extracts to alter the resistance towards oxidative stress. For this purpose, we used paraquat, which is a known oxidative stress inducer and greatly decreases the lifespan of *C*. *elegans* [70,71]. In this assay, both *C*. *chinensis* and *E*. *ulmoides* extracts increased the mean survival of paraquat-treated aged nematodes. There are two conceivable mechanisms, which could have led to the increased survival during oxidative stress. The first possibility is an enhanced prevention of oxidative stress by antioxidative mechanisms, so that the ROS abundance is lower in extract-treated nematodes. The second possibility is rather a “curative” approach by inducing repair mechanisms, which enables cells to better cope with the consequences of stress. We recently found that the chaperones *hsp*-*16*.*1* and, to some extent, *hsp*-*12*.*6*, are upregulated in *C*. *chinensis*-treated nematodes on the 12th day of adulthood [16]. These heat shock proteins could help the cells cope with stress by protecting intracellular proteins from misfolding or aggregation [72]. Interestingly, *E*. *ulmoides*-treated nematodes did not show those upregulations, but only five heat shock proteins were covered in this qPCR assay.

To address the question whether the TCM extracts can reduce intracellular ROS, and thus prevent oxidative stress, the ROS level was evaluated in aged *C*. *elegans* with the aid of the HyPer-transgenic strain. We showed that both extracts lead to a reduction of ROS. This finding corroborates several studies which already described the role of *C*. *chinensis* and *E*. *ulmoides* as antioxidant agents in other model organisms [73,74,75,76,77,78,79]. The antioxidant activity of *C*. *chinensis* may be attributed to the ability to increase the levels of antioxidant enzymes, especially superoxide dismutase, catalase and glutathione peroxidase [73,79].

However, the implications of the HyPer-assay are limited. First, the worm fluorescence images did not exhibit a black background due to the long exposure-time, potentially leading to interferences. Second, whole-animal fluorescence measurements might be not optimal. Back et al. [36] demonstrated that the fluorescence of the HyPer-strain is tissue dependent (especially in young adult worms); for instance, ROS level was low in the gonads and intestine yet relatively high in muscle cells, certain neurons, and the hypodermis. Therefore, intracellular ROS measurements in certain cells could provide more accurate information than whole body measurements. With the improved mechanosensory response and swimming behaviour after the *C*. *chinensis* treatment, the ROS level in mechanosensory neurons and body wall muscle cells could potentially present a direct link from the antioxidative capacity of *C*. *chinensis* to certain observed health improvements. Third, the increased ROS level after the extract exposure in the H_2_DCFDA-assay cannot be completely ignored, despite the shortcomings presented (see Appendix A).

Undoubtedly, aged nematodes pre-treated with the *C*. *chinensis* or *E*. *ulmoides* extract showed a prolonged survival during the exogeneous oxidative stress exposure. However, it remains unclear if the reduced intracellular ROS level (as shown in the HyPer assay) or, conversely, the increased ROS level (as shown in the H_2_DCFDA-assay) is the underlying cause for the protection; the latter could lead to a protection against exogeneous oxidative stress in accordance with the (mito)hormesis hypothesis [80,81]. Further methods to quantify the endogenous ROS level inside the aged nematodes after extract treatments are necessary to uncover the true mechanisms.

### 4.3. A Hydroxygallic Acid Derivative and a Sterol Lipid Are Potential Triggers for the C. chinensis-Induced Increase of Physical and Cognitive Fitness

We wanted to further understand why the *C*. *chinensis* and *E*. *ulmoides* extracts elicit such differing responses in *C*. *elegans*: *C*. *chinensis*, as an overall healthspan enhancer, improves the physical, physiological [16] and cognitive fitness; while *E*. *ulmoides* is only able to enhance the physiological fitness of *C*. *elegans* after stress. Therefore, we needed to examine the extract compounds.

We analyzed the constituents present in the organic extract of *C*. *chinensis* and compared them to those in the *E*. *ulmoides* extract using UPLC-MS. A hydroxygallic acid derivative was highly abundant in the *C*. *chinensis* extract, and represented more than 5% of the total peak area in the crude organic extract. Hydroxygallic acid was previously isolated from pomegranate peels [82], clove (*Syzygium aromaticum*) [83] and *Gymnocarpos decandrus* Forssk [84]. However, scientific reports about bioactivities are rare. Most studies focused on the closely related gallic acid, which was shown to boost the health- and lifespan of *C*. *elegans* [85], and which offers numerous therapeutic applications [86]. Abundance does not necessarily equal importance; however, it is interesting that the hydroxygallic acid derivative was relatively sparse in the *E*. *ulmoides* extract, but highly abundant in the *C*. *chinensis* extract and its most active fraction (O). Thus, it is conceivable that the hydroxygallic acid derivative is (at least partly) responsible for the increase of physical and cognitive fitness caused by *C*. *chinensis*.

In this regard, the sterol lipid “4-alpha-formyl-stigmasta-7,24(241)-dien-3-beta-ol” is also worth mentioning, whose structure is based on the stigmastane skeleton. It was one of the highly abundant compounds documented in the *C*. *chinensis* extract, and was not detected in the *E*. *ulmoides* organic extract at all. However, in *C*. *chinensis* fraction O, it only occurred in trace amounts. It is found in several plants, such as highbush blueberry (*Vaccinium corymbosum*), blackberry (*Rubus fruticosus*), raspberry (*Rubus ideaeus*), nopal (*Opuntia ficus*-*indica*), and oil-seed camellia (*Camellia oleifera*) (see HMDB0304197 in the Human Metabolome Database; https://hmdb.ca/metabolites/ accessed on 4 September 2022). No bioactivities were reported so far. Thus, both hydroxygallic acid derivatives and 4-alpha-formyl-stigmasta-7,24(241)-dien-3-beta-ol should be studied in bioassays in detail to uncover their potential healthspan-extending abilities.

### 4.4. Several Bioactive Compounds Could Be the Underlying Cause of Enhanced Healthspan during C. chinensis and E. ulmoides Treatment

Cinnamic acid, an aromatic carboxylic acid, was also one of the highly abundant compounds in the *C*. *chinensis* extract, but had a tenfold lower abundance in the *E*. *ulmoides* extract. It is found in several plants such as *Cinnamomum cassia* [87], *Allium fistulosum* [88], *Ocimum gratissimum*, as well as *Vitellaria paradoxa* [89]. Furthermore, it represents a central intermediate in the biosynthesis of numerous phytochemical compounds, including coumaric and chlorogenic acids [90], ferulic acid and curcumin [91], as well as caffeic acid and p-hydroxycinnamic acid [92]. Cinnamic acid derivatives were also shown to exhibit antibacterial [88], antidiabetic [93], anti-inflammatory, antimicrobial [94], antioxidant [95] and anticancer [96] bioactivities.

Other bioactive compounds found in the *C*. *chinensis* and *E*. *ulmoides* extracts were coumarin, chlorogenic acid, kaempferol, caffeoylquinic acid derivatives, quercetin, and isorhamnetin. Coumarin features antiparasitic, anti-inflammatory, and antidiarrheal activities [97,98]; Amari et al. [99] reported that the extract of *Thymelaea hirsuta* has antifungal and antiaging activities caused by coumarin derivatives. Chlorogenic acid was found to have anti-inflammatory and antiviral effects [100] as well as anti-oxidative and neuroprotective activities in several studies [101,102,103,104]. Choe et al. [105] reported that kaempferol isolated from the roots of *Rhodiola sachalinensis* features anti-inflammatory and antioxidant activities. Moreover, anti-proliferative and proapoptotic activities of kaempferol against various types of cancers were reported, including breast [106], lung [107], colon [108], and bladder [109]. Caffeoylquinic acid derivatives, which are more abundant in *E*. *ulmoides* than in *C*. *chinensis*, have shown anti-inflammatory, antioxidant [110], and neuroprotective effects [111].

Quercetin is probably one of the most interesting polyphenols in the human diet with high abundance in various fruits and vegetables [112]. Recently, it gained attention for many pharmacological actions, such as neuroprotective effects [113]; improvement of cognitive disorders in aging mice [114]; as well as antioxidant [115], antimicrobial [116,117], and anticancer activity [118]. Furthermore, its healthspan- and lifespan-promoting abilities in *C*. *elegans* [119,120,121] make it a suitable candidate as one of the healthspan-promoting compounds in both extracts, although quercetin and its derivatives are more abundant in the *E*. *ulmoides* extract. Finally, isorhamnetin, a methylated flavonol and a derivative of quercetin, is found in blackberries, apples, pears, cherries and several medicinal herbs [122,123]. Isorhamnetin has many beneficial properties, including antioxidant, anti-tuberculosis, anti-inflammatory, antimicrobial, anti-obesity, anticancer, hepatoprotective, and antidiabetic effects [124,125].

Ultimately, several bioactive molecules could be involved in the beneficial effects of the tested extracts. Moreover, their interaction may underlie the observed effects in additive or even synergistic ways, rather than one single compound. This assumption is supported by several studies, reviewed in [126,127,128]. Thus, further studies on bioactivities of single constituents, and especially their combinations, are necessary to find the underlying cause for healthspan benefits. Here, we started this challenging approach by studying three fractions enriched in single compounds of *C*. *chinensis*, discussed in the following section.

### 4.5. Zingibroside R1 Mirrors the Beneficial Actions of C. chinensis

In our study, we obtained three fractions enriched in astragalin, pinoresinol, and zingibroside R1 from the *C*. *chinensis* seed extract by a preparative HPLC method, and analyzed their bioactivity. Structures of compounds were determined by a combination of MS and NMR spectroscopy.

Astragalin (kaempferol-3-O-β-D-glucoside) is a naturally occurring flavonoid that has been isolated and identified in several plants [129], such as *Prunus persica* [130], *Lespedeza cuneata* [131], *Moringa oleifera* Lam [132] *Astragalus membranaceus* root [133], *Allium ursinum* [134], *C*. *chinensis* and *Cuscuta australis* [135]. Multiple pharmacological effects of astragalin have been reported, such as anti-inflammatory [136,137], antioxidant [132], anti-apoptotic [138], antimicrobial [134], anti-osteoporotic [139], anti-anticancer [140,141], as well as neuro- and cardio-protective [142] activities. *In vivo*, the glucoside may be hydrolyzed readily to yield the aglycone kaempferol [143], which may be responsible for (part of) its effects.

Among the three fractions enriched in single compounds tested in this study, the fraction enriched in astragalin showed the highest ability to improve resistance against heat stress; however, the *C*. *chinensis* extract remains the strongest modulator. Furthermore, the fraction enriched in astragalin increased lifespan in a similar way to the extract, but showed little effect in the locomotion assay. Thus, it can only partially account for the beneficial effects of *C*. *chinensis*. This life-prolonging effect in *C*. *elegans* was also reported for astragalin isolated from *Radix tetrastigma* [144].

Pinoresinol is a furfuran-type lignan found in several plants such as *E*. *ulmoides* [145,146], *Sambucus williamsii* [147], *Brassica* vegetables [148], legumes [149], as well as sesame seed and olive oil [150]. Pinoresinol has multiple pharmacological effects, including antioxidant, anti-inflammatory, and anticancer activities [145,151,152]. Moreover, it can decrease neuro-inflammation, apoptosis and memory impairment [145,153]. The fraction enriched in pinoresinol slightly improved the resistance of *C*. *elegans* towards heat shock, but did not achieve improvement in either lifespan or swimming behavior. Koch et al. [154] showed that the incubation of *C*. *elegans* with 100 µM pinoresinol did not affect lifespan, ROS accumulation, or thermal resistance in *C*. *elegans*, but did induce the nuclear translocation of the transcription factor DAF-16, which plays a key role in the health-relevant insulin/IGF-like signaling pathway. The contradictory observations regarding the potential of pinoresinol to increase stress resistance may be due to the usage of different concentrations. Furthermore, Koch and colleagues tested the stress resistance on the second day of adulthood, whereas we used 12-day-old worms, which is hardly comparable [155]. Due to its weak performance in the selected assays, pinoresinol is assumed to play a rather negligible role in the healthspan-promoting effects of *C*. *chinensis*.

Zingibroside R1 belongs to the triterpenoid oleanane-type saponins, and occurs naturally as a secondary metabolite in several plants, especially those belonging to the family of *Araliaceae* and *Panax* species [156,157]. It was identified in the rhizomes, taproots, and lateral roots of *Panax japonicas* [158,159], the rhizomes of *Panax zingiberensis* [158], the roots of *Achyranthes bidentata* [160], and the roots and rhizomes of *Panax ginseng* [156,161]. Zingibrosides exhibit anti-tumor and anti-angiogenic activities as well as inhibitory effects against HIV-1 [158,162]. In our study, the fraction enriched in zingibroside R1 enhanced lifespan and locomotion, and slightly increased the mean survival of the nematodes after heat stress.

To summarize, all three fractions enriched in single compounds enhanced the survival of *C*. *elegans* after the exposure to heat stress on the 12th day of adulthood. Lifespan without stress exposure was improved by both the fractions enriched in astragalin and zingibroside R1, while locomotion was only enhanced by the fraction enriched in zingibroside R1. Thus, zingibroside R1 comes closest in mimicking the effects of the *C*. *chinensis* extract for the healthspan benefits observed in *C*. *elegans*. This does not preclude the possibility that astragalin, pinoresinol or other (yet to be identified) compounds play a role in the overall activity of the *C*. *chinensis* extract via additive or synergistic effects.

## 5. Conclusions

This study demonstrates the neuroprotective abilities of *C*. *chinensis* by improving memory and mechanosensation in aged nematodes, as well as oxidative stress resistance. In contrast, the *E*. *ulmoides* extract only increased oxidative stress resistance, which is in line with our past report [16]. Antioxidative capacities of both extracts might be one of the underlying causes of the healthspan benefits. Based on chemical analyzes of the extracts by UPLC-MS/MS, we conclude that hydroxygallic acid derivatives and the sterol lipid 4-alpha-formyl-stigmasta-7,24(241)-dien-3-beta-ol are promising candidates for the specific health effects of *C*. *chinensis*. Both substances are highly abundant in the *C*. *chinensis* extract and in one of its most effective fractions, but are much less present in the *E*. *ulmoides* extract. Furthermore, several bioactive compounds were identified in *C*. *chinensis* and *E*. *ulmoides*, which may act in an additive or synergistic manner to increase life- and healthspan. A fraction from the *C*. *chinensis* extract enriched in one of these compounds, zingibroside R1, has the ability to extend lifespan, enhance heat stress resistance, and improve the locomotion of *C*. *elegans* on the 12th day of adulthood in liquid media. Collectively, our results are evidence for an overall anti-aging effect of *C*. *chinensis* in *C*. *elegans* and provide the first hints regarding the compounds responsible for these observations.

## Figures and Tables

**Figure 1 nutrients-14-04199-f001:**
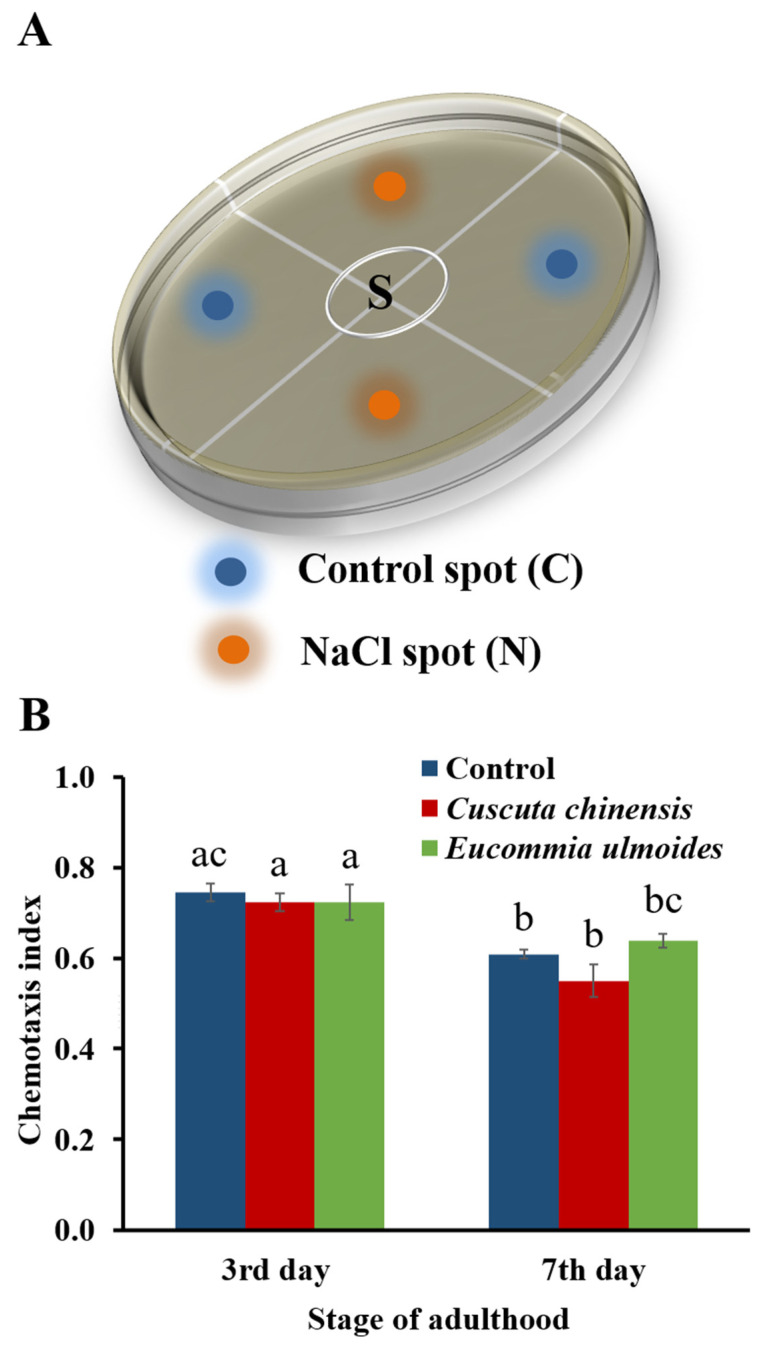
Chemotaxis assays of *C. elegans* treated with *Cuscuta chinensis* and *Eucommia ulmoides* extracts. (**A**) The design of the chemotaxis assay plates is shown, whereby “S” marks the starting spot of the nematodes, as well as (**B**) the chemotaxis indices on the 3rd and 7th day of adulthood. Each test was repeated twice with 6 plates and *n* ≥ 400 nematodes per treatment in total. The bars represent the mean ± SEM and different letters above the bars indicate a significant difference (*p* < 0.05) according to a one-way ANOVA and post-hoc Bonferroni test.

**Figure 2 nutrients-14-04199-f002:**
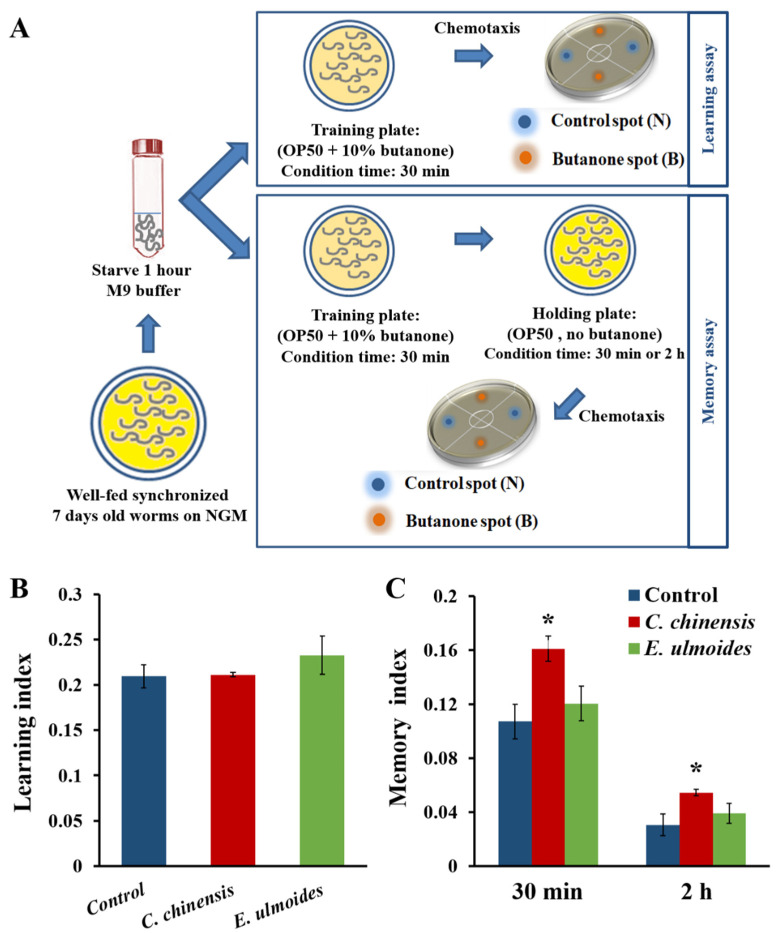
Impact of *C. chinensis* and *E. ulmoides* extract on the ability of *C. elegans* to learn and remember the association between food and butanone on the 7th day of adulthood. (**A**) The diagram gives an overview of the associative olfactory learning and memory assays in *C. elegans*. (**B**) The mean learning indices ± SEM of *n* ≥ 400 nematodes and (**C**) the mean STAM indices after two different holding periods (30 min and 2 h) ± SEM of *n* ≥ 400 nematodes per treatment is shown. Differences are considered significant with * (*p* < 0.05), according to a one-way ANOVA and post-hoc-Bonferroni test.

**Figure 3 nutrients-14-04199-f003:**
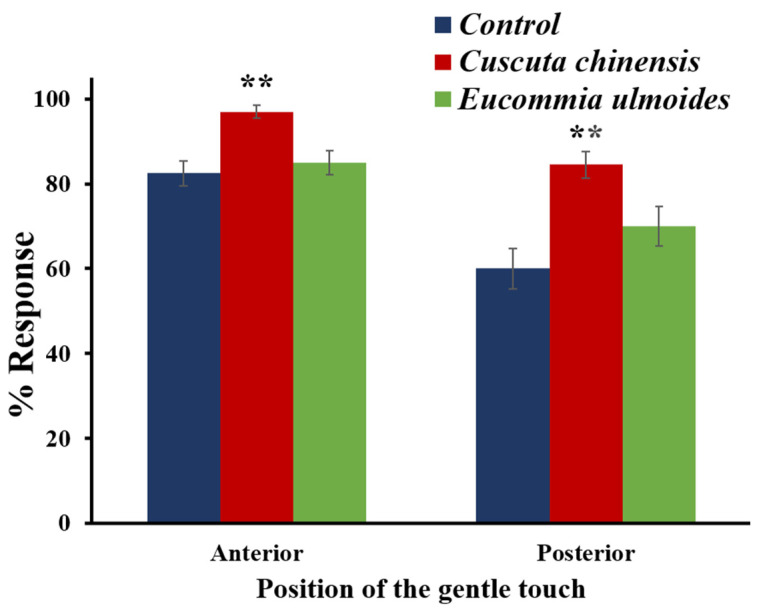
Impact of *C. chinensis* and *E. ulmoides* extract on the mechanosensory response of *C. elegans* to gentle touches on the 12th day of adulthood. The rate of behavioral responses of *C. chinensis* and *E. ulmoides*-treated worms to anterior and posterior touches were recorded for *n* ≥ 40 nematodes per treatment. The scores were represented as mean % ± SEM and significant differences to the control are considered with ** (*p* < 0.01) according to a one-way ANOVA and post-hoc Bonferroni test.

**Figure 4 nutrients-14-04199-f004:**
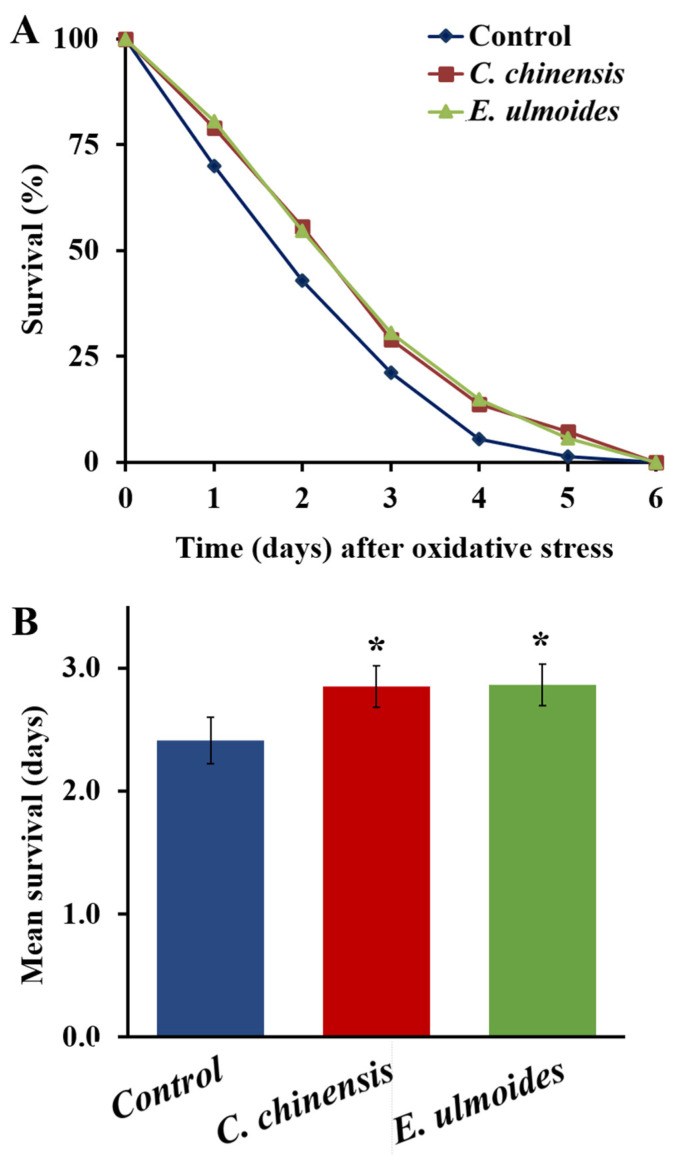
Augmenting the oxidative stress resistance of *C. elegans* treated with *C. chinensis* and *E. ulmoides*. (**A**) The survival curves of *C. elegans* treated with *C. chinensis* and *E. ulmoides* since L4 stage, during the exposure to oxidative stress (60 mM paraquat) starting on the 12th day of adulthood are shown. (**B**) The mean survival ± SEM from three biological replicates is shown in addition. Significant differences were determined by a log-rank test and subsequent Bonferroni correction with * *p* < 0.05.

**Figure 5 nutrients-14-04199-f005:**
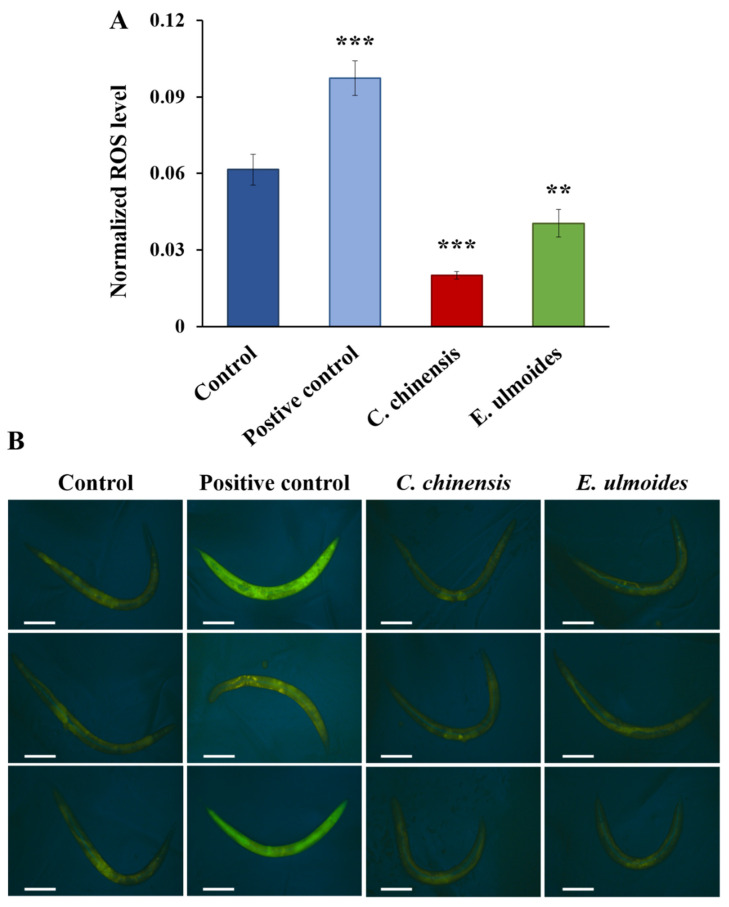
ROS level in *C. elegans* on the 12th day of adulthood treated with *C. chinensis* and *E. ulmoides*. (**A**) The mean normalized fluorescence intensity is shown for the HyPer transgenic strain on the 12th day of adulthood. Error bars represent the standard error of the mean (SEM) and significant differences to the control are considered with ** (*p* < 0.01) or *** (*p* < 0.001), according to a one-way ANOVA and post-hoc Bonferroni test. (**B**) Representative images for the fluorescence intensity of the HyPer transgenic strain treated with DMSO (control), 10 mM H_2_O_2_ (positive control), *C. chinensis* and *E. ulmoides* extract are shown. The white scale bars represent 200 µm.

**Figure 6 nutrients-14-04199-f006:**
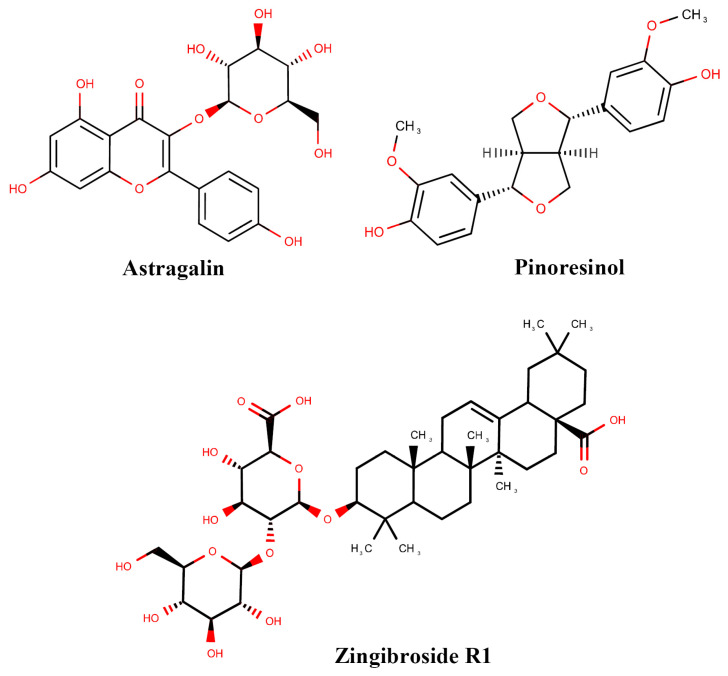
The chemical structure of the compounds (astragalin, pinoresinol, and zingibroside R1) from the seeds of *C*. *chinensis*.

**Figure 7 nutrients-14-04199-f007:**
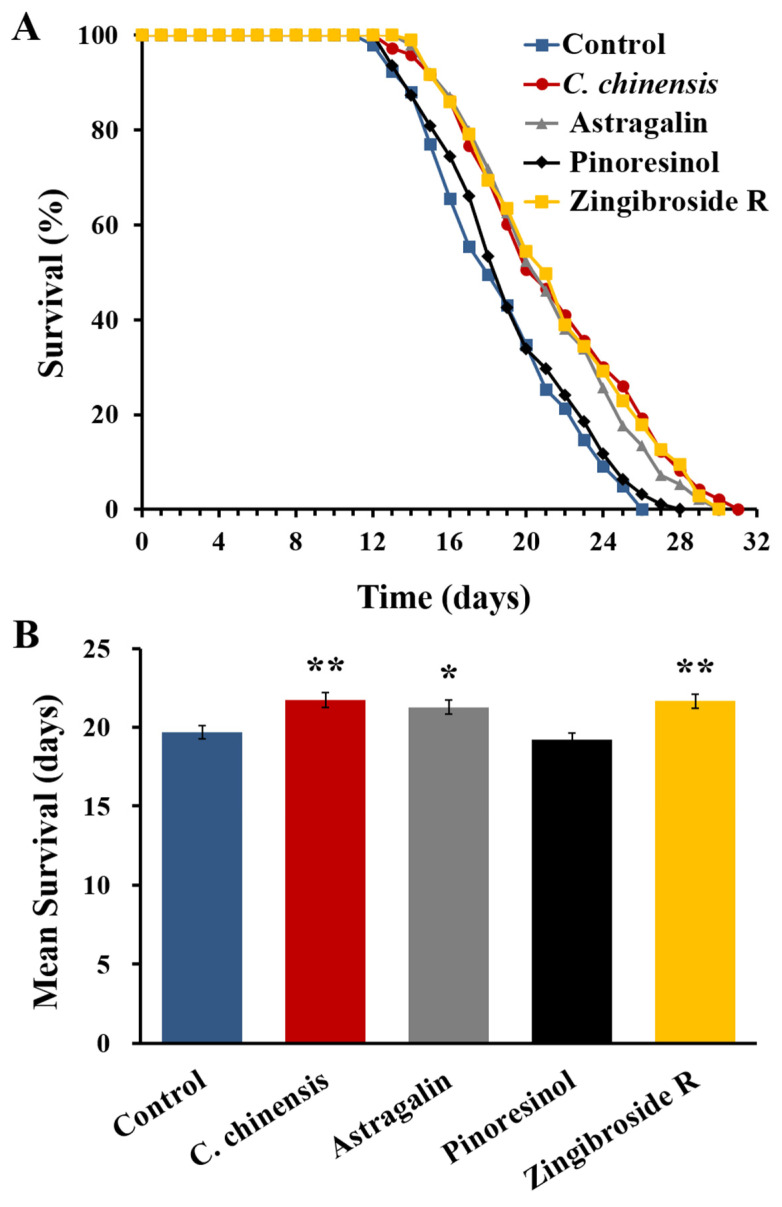
Effect of the *C. chinensis* extract and its fractions enriched in astragalin, pinoresinol, and zingibroside R1 on the lifespan of *C. elegans*. (**A**) Representative survival curves and (**B**) mean lifespan ± SEM from three biological replicates. Significant differences were determined by a log-rank test and subsequent Bonferroni correction with * *p* < 0.05 or ** *p* < 0.01.

**Figure 8 nutrients-14-04199-f008:**
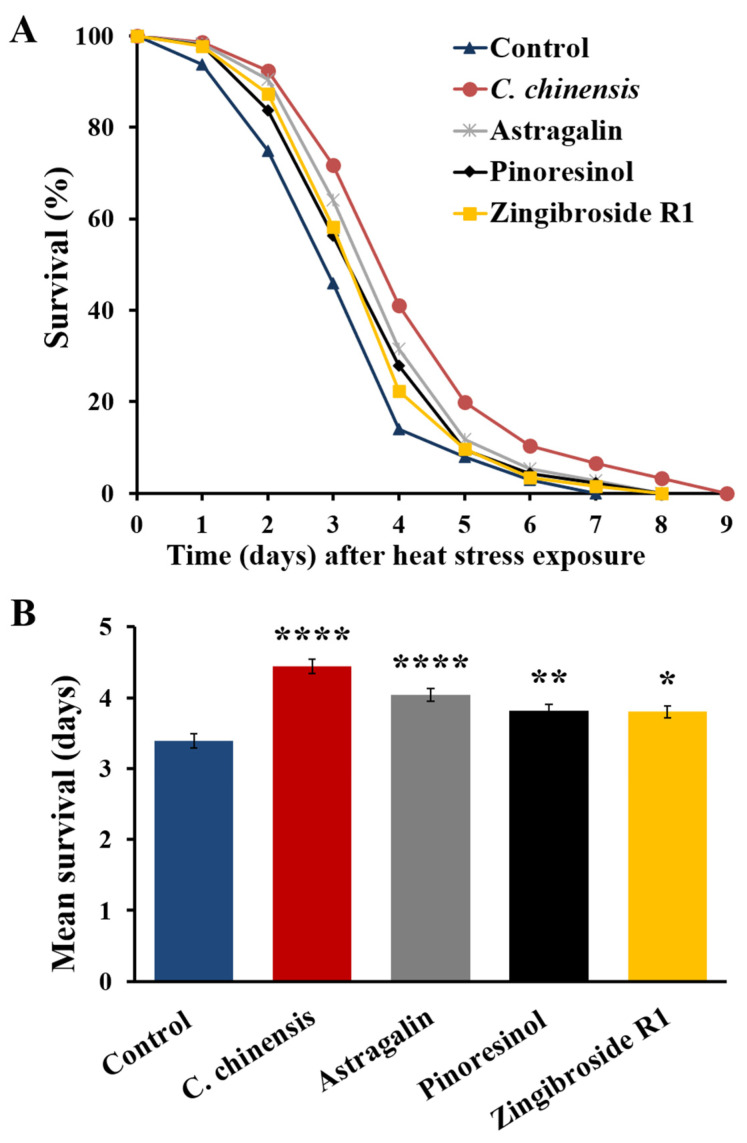
Survival of *C. elegans* treated with *C. chinensis* and its fractions enriched in astragalin, pinoresinol, and zingibroside R1 after heat stress. (**A**) Representative survival curves of *C. elegans* following heat stress exposure (37 °C for 3 h) on the 12th day of adulthood and (**B**) mean survival ± SEM after stress exposure from three biological replicates. Significant differences were determined by a log-rank test and Bonferroni correction with * *p* < 0.05; ** *p* < 0.01; and **** *p* < 0.0001.

**Figure 9 nutrients-14-04199-f009:**
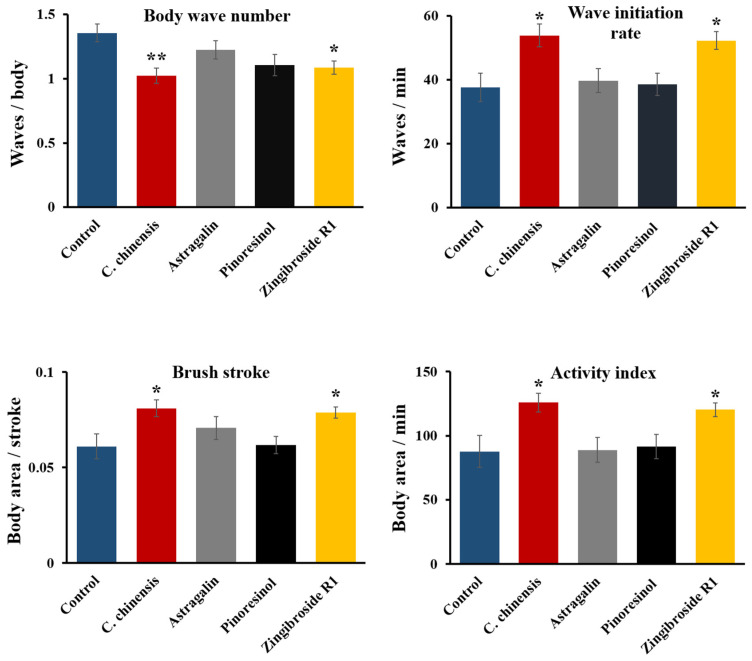
Swim performance of *C. elegans* after treatment with *C. chinensis* and its fractions enriched in astragalin, pinoresinol, and zingibroside R1. Body wave number, wave initiation rate, brush stroke, and activity index were determined on the 12th day of adulthood. Error bars are the standard error of the mean (SEM) and one bar represents *n* ≥ 50 from two independent trials. Statistical significance was determined according to a one-way ANOVA and post-hoc Bonferroni test with * (*p* < 0.05) and ** (*p* < 0.01).

**Table 1 nutrients-14-04199-t001:** Effect of *C. chinensis* and *E. ulmoides* extract on the survival of *C. elegans* during exposure to paraquat, initiated on the 12th day of adulthood.

Treatment	Mean Survival		Min. Survival [Days]	Med. Survival [Days]	Max. Survival [Days]	*n*	*p*-Value (Compared to Control)
	Days ± SE	%					
Control	2.41 ± 0.19	100	0.84	2.19	6	147	
*C. chinensis*	2.85 ± 0.17	118.3	1.17	3.71	6	124	0.0132
*E. ulmoides*	2.86 ± 0.17	118.7	1.21	3.43	6	108	0.0144

*n*, total number of worms; SE, Standard error; Min/Med/Max, minimum/median/maximum survival (days until deaths in population reached 25/50/100%). The *p*-values were calculated using a log-rank test with subsequent Bonferroni correction.

**Table 2 nutrients-14-04199-t002:** Most abundant compounds in the extract from the seeds of *C. chinensis,* in comparison with its fraction O and the *E. ulmoides* extract.

Compound	*m*/*z* (Da)	RT (min)	*C. chinensis*	*E. ulmoides*	Fraction O
			Peak Area (%)	Peak Area (%)	Peak Area (%)
Hydroxygallic acid derivative	187.096	8.002	5.02	0.67	4.70
Unidentified	195.308	0.630	4.50	0.60	0.01
2-C-Methyl-D-erythritol 4-phosphate	215.032	0.643	2.15	1.02	3.5 × 10^−3^
Dienoic acid derivative	311.222	14.082	2.10	2.1 × 10^−3^	7.0 × 10^−3^
d-Pinoresinol-4-O-glucoside	519.186	7.558	1.23	0.13	0.01
Stearic acid or hexadecenoic acid	311.201	14.806	1.15	0.12	0.03
Cinnamic acid derivative	265.147	14.665	1.14	0.17	0.07
Coumarin derivative	147.044	7.373	1.12	1.8 × 10^−3^	2.3 × 10^−4^
Hexadecanoic acid	315.253	13.870	1.07	0.09	0.35
Disaccharide	341.108	0.707	1.05	0.29	0.03
Disaccharide	377.085	0.708	1.05	0.46	4.2 × 10^−3^
Disaccharide	387.114	0.707	0.85	0.29	0.03
Palmitic acid or hexadecanoate	315.253	13.720	0.94	0.05	0.20
Unidentified	187.006	4.116	0.91	0.01	2.2 × 10^−5^
4-alpha-formyl-stigmasta-7,24(241)-dien-3-beta-ol	455.352	15.514	0.83	0	4.7 × 10^−3^
Oleic acid derivative	339.232	15.544	1.00	2.62	4.0 × 10^−4^
Zingibroside R1	793.437	12.132	0.07	0.02	0.02
Genistein derivative	431.228	13.926	0.04	2.0 × 10^−4^	3.2 × 10^−4^
Chlorogenic acid	353.087	4.978	0.03	0.12	3.0 × 10^−3^

*m*/*z*: mass to charge ratio, RT: retention time (min), Da: Dalton.

**Table 3 nutrients-14-04199-t003:** Characterization of bioactive compounds in extracts of the seeds of *C. chinensis*, its fraction O and the bark of *E. ulmoides.*

Compound	*m*/*z* (Da)	RT (min)	*C. chinensis*	*E. ulmoides*	Fraction O
			Peak Area (%)	Peak Area (%)	Peak Area (%)
Apigenin	269.045	10.484	5.5 × 10^−4^	0.03	3.3 × 10^−3^
Astragalin	447.092	8.269	5.1 × 10^−5^	3.8 × 10^−4^	2.4 × 10^−4^
Caffeoylquinic acid derivative	515.139	5.473	5.3 × 10^−4^	0.30	1.0 × 10^−3^
Caffeoylquinic acid derivative	515.140	4.958			
Caffeoylquinic acid derivative	515.140	4.898			
Caffeoylquinic acid derivative	515.136	5.402			
Caffeoylquinic acid derivative	515.139	5.183			
Caffeoylquinic acid derivative	515.139	4.602			
Caffeoylquinic acid derivative	515.139	4.832			
Caffeoylquinic acid derivative	515.140	4.339			
Caffeoylquinic acid derivative	515.140	3.576			
Caffeoylquinic acid derivative	515.140	4.029			
Caffeoylquinic acid derivative	515.140	3.678			
Chlorogenic acid derivative	353.087	5.584	0.02	0.08	9.1 × 10^−4^
Chlorogenic acid derivative	353.087	5.120			
Isorhamnetin-o-glucoside	477.103	7.783	0	0.20	2.8 × 10^−3^
Isorhamnetin-o-glucoside	477.103	7.675			
Kaempferol 3-O-rhamnoside-7-O-glucoside	593.148	5.787	3.9 × 10^−3^	0	5.2 × 10^−5^
Kaempferol 3-O-rhamnoside-7-O-glucoside	593.149	0.738			
Kaempferol-3-rhamnosyhexose	593.129	9.449	0	0.12	2.3 × 10^−3^
Kaempferol-glycosides derivative	593.129	9.335	0	0.15	0.01
Kaempferol-glycosides derivative	755.182	7.402			
Kaempferol-glycosides derivative	755.181	7.663			
Kaempferol-o-coumaroyglucoside-o-hexoside	755.182	7.941	0	0.01	3.7 × 10^−3^
Kaempferol-O-dihexoside	609.145	5.505	0	0.07	3.7 × 10^−3^
Lignan-coumaroylglucoside	697.364	11.582	5.1 × 10^−4^	0.26	0.02
Lignan-coumaroylglucoside derivative	697.364	11.440			
Lignan-coumaroylglucoside derivative	697.364	11.141			
Lignan-o-coumaroylglucoside derivative	683.349	10.991			
Quercetin	301.035	7.002	1.6 × 10^−4^	0.03	2.0 × 10^−4^
Quercetin-apiosyl-galactose	595.129	6.579	0	1.07	3.5 × 10^−3^
Quercetin-glucosides derivative	625.141	6.702	2.6 × 10^−5^	1.08	3.5 × 10^−3^
Quercetin-glucosides derivative	625.140	5.970			
Quercetin-glucosides derivative	625.141	6.702			
Quercetin-glucosides derivative	625.141	6.910			
Rutin	609.145	6.814	3.1 × 10^−4^	0.01	1.5 × 10^−4^
Secoisolariciresinol	361.165	12.401	3.8 × 10^−4^	1.0 × 10^−4^	1.8 × 10^−4^

*m*/*z*: mass to charge ratio, RT: retention time (min), Da: Dalton.

**Table 4 nutrients-14-04199-t004:** Effect of the *C. chinensis* extract and its fractions enriched in astragalin, pinoresinol, and zingibroside R1 on the lifespan of *C. elegans.*

Treatment	Mean Lifespan		Min. [Days]	Med. [Days]	Max. [Days]	*n*	*p*-Value (Compared to Control)
	Days ± SE	%					
Control	19.69 ± 0.42	100.0	16.0	18.8	28.0	91	
*C. chinensis*	21.74 ± 0.47	110.4	17.4	20.8	31.0	98	0.001
Astragalin	21.28 ± 0.43	108.1	17.4	20.2	30.0	98	0.003
Pinoresinol	19.23 ± 0.40	97.7	15.9	18.3	28.0	93	1.000
Zingibroside R1	21.66 ± 0.46	110.0	17.5	21.0	30.0	94	0.041

*n*, total number of worms; SE, Standard error; Min/Med/Max, minimum/median/maximum lifespan (days until deaths in population reached 25%/50%/100%). The *p*-values were determined by using a log-rank test with subsequent Bonferroni correction.

**Table 5 nutrients-14-04199-t005:** Effect of the *C. chinensis* extract and its fractions enriched in astragalin, pinoresinol, and zingibroside R1 on the survival of *C. elegans* after heat stress exposure (37 °C for 3 h) on the 12th day of adulthood.

Treatment	Mean Lifespan		Min. [Days]	Med. [Days]	Max. [Days]	*n*	*p*-Value (Compared to Control)
	Days ± SE	%					
Control	3.39 ± 0.10	100.0	1.99	2.86	7.00	179	
*C. chinensis*	4.44 ± 0.11	131.0	2.84	3.71	9.00	212	<0.0001
Astragalin	4.04 ± 0.09	119.2	2.59	3.43	8.00	229	<0.0001
Pinoresinol	3.82 ± 0.09	112.7	2.32	3.22	8.00	233	0.0098
Zingibroside R1	3.80 ± 0.08	112.1	2.42	3.23	8.00	261	0.0137

*n*, total number of worms; SE, Standard error; Min/Med/Max, minimum/median/maximum lifespan (days until deaths in population reached 25%/50%/100%). The *p*-values were determined by using a log-rank test with subsequent Bonferroni correction.

## Data Availability

The data presented in this study are included in this published article and its Appendix A.

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
