# Peer review of "Identification of a Hydroxygallic Acid Derivative, Zingibroside R1 and a Sterol Lipid as Potential Active Ingredients of Cuscuta chinensis Extract That Has Neuroprotective and Antioxidant Effects in Aged Caenorhabditis elegans"

_nutrients, 2022, doi:10.3390/nu14194199_

Round 1

Reviewer 1 Report

The manuscript entitled “Identification of a hydroxygallic acid derivative, zingibroside R1 and a sterol lipid as potential active ingredients of Cuscuta chinensis extract that has neuroprotective and antioxidant effects in aged Caenorhabditis elegans”, is a good scientific paper with interesting results and useful methods, so I wish to congratulate the authors for their work. However, there are some issues that should be addressed prior publication.

1-     English should be revised in the manuscript

2-      Section 2.1, I understand that after the fractionation of the sample either by RP-HPLC or MPLC the samples are diluted in mobile phase at pH 3.0, ¿was the organic solvents evaporated before used in the in vivo assays? ¿Was the pH adjusted before used in the in vivo assays?, if not please explain why. Please clarify these questions in the corresponding section.

3-     Section 2.1 Please include the apparatus used (model, manufacturer...) also include the detection/monitoring method used ELSD, PDA, UV as well as the operating details, temperature, wavelength…

4-     Section 2.1 L 100. What is the meaning of: “The single substances astragalin, pinoresinol and zingibroside R1 were enriched from C. chinensis”, were the molecules isolated from the extract? Were they purified? Which was the purity grade obtained?. Usually, an extract is enriched in a determined molecule, however an isolated molecule could not be enriched only purified from the extract. Please made the necessary changes along the manuscript

5-     Please improve the quality of the Figures S6-S8

6-     L 100 molecule or compound or phytochemical instead of single substance

7-     L 148 and 151 ml should be mL, please ensure the homogeneity along the manuscript

8-     Section 2.9. Include the characteristics of the used microscope also the magnification used. Please explain how the autofluorescence was subtracted, also the analysis method and software used for it. Change  NaN3 for sodium azide

9-     L 241 experiments instead of repeats

10- Section 2.10, is not clear if the experiment plates were supplemented with OP50, please explain

11- L 244 differences instead of changes

12- Figure 1, the authors stated in L 253 that chemotaxis index fades with aging, therefore were the differences between the two days assayed significant, i.e the difference between the CI at 3rd day and CI at 7th day for the control animals is significant

13- Figure 5, scale bars should appear in the images, moreover images seem to be overexposed, the background in fluorescent images must be black to avoid interferences. A representative image of a treated worm should be included.

14- L 349 image instead of picture

15- L 513 improve or extend instead of change

16- L 518 please rewrite the sentence: the most efficient survival improvement

17- L 646 survival instead of survivability

18- L 647 compounds instead ingredients

19- L664-665 please include the scientific names of the mentioned plants

20- Appendix A could be included in supplementary material.

21- L 834 Subtracting the autofluorescence of the controls worms is not the best method to avoid interferences. Splitting the images in RGB and subtracting the red channel to the green channel is a better solution, using lower exposure times works fine too.

22- L 856 Indeed the H2DCFDA method is not specific for H2O2, however there are selective probes to quantify superoxide such as hydroethidine

23- Figure A1 scale bars should appear in the images. A representative image of a treated worm should be included.

Author Response

Dear Ms. Ding, dear Reviewers,

We thank you very much for the constructive suggestions! We tried our best to fulfill all requests. Please find our comments targeting all mentioned issues below:

Reviewer 1

  • English should be revised in the manuscript

We addressed this issue by a native speaker, who revised the language in the whole manuscript and who is acknowledged in the acknowledgement section. Several language improvements were made, especially in the discussion-section.  

  • Section 2.1, I understand that after the fractionation of the sample either by RP-HPLC or MPLC the samples are diluted in mobile phase at pH 3.0, ¿was the organic solvents evaporated before used in the in vivo assays? ¿Was the pH adjusted before used in the in vivo assays?, if not please explain why. Please clarify these questions in the corresponding section.

The samples were completely dried and freshly dissolved prior to the assays. Thereafter, they were diluted with assay-buffer. This information was added to lines 101-102.

  • Section 2.1 Please include the apparatus used (model, manufacturer...) also include the detection/monitoring method used ELSD, PDA, UV as well as the operating details, temperature, wavelength…

As suggested, we added apparatus and methodological details in section 2.1 (line 121-130).

4-     Section 2.1 L 100. What is the meaning of: “The single substances astragalin, pinoresinol and zingibroside R1 were enriched from C. chinensis”, were the molecules isolated from the extract? Were they purified? Which was the purity grade obtained?. Usually, an extract is enriched in a determined molecule, however an isolated molecule could not be enriched only purified from the extract. Please made the necessary changes along the manuscript

We changed the choice of words according to your suggestion throughout the manuscript. In addition, we added a more detailed description about the used methodology and the purity grade (line 109-121).

5-     Please improve the quality of the Figures S6-S8

We improved the quality of the figures and split the LCMS and NMR spectra to separated figures (Figures S6-S11).

6-     L 100 molecule or compound or phytochemical instead of single substance

It was changed as suggested (line 109).

7-     L 148 and 151 ml should be mL, please ensure the homogeneity along the manuscript

The manuscript was checked for the correct usage of “µL” and “mL” and several corrections were made.

8-     Section 2.9. Include the characteristics of the used microscope also the magnification used. Please explain how the autofluorescence was subtracted, also the analysis method and software used for it. Change  NaN3 for sodium azide

We added the information according to your suggestions (line 241-250).

9-     L 241 experiments instead of repeats

We changed it as suggested (line 250).

10- Section 2.10, is not clear if the experiment plates were supplemented with OP50, please explain

Yes, OP50 was also used in the reproduction assay. The information was added in line 256.

11- L 244 differences instead of changes

We changed it as suggested (line 281).

12- Figure 1, the authors stated in L 253 that chemotaxis index fades with aging, therefore were the differences between the two days assayed significant, i.e the difference between the CI at 3rd day and CI at 7th day for the control animals is significant

Yes, there is a significantly difference between CI on the 3rd day and 7th day of adulthood in the treatment groups. We now included this information in Figure 1 by using a letter code instead of stars and also included it in lines 291-292.

13- Figure 5, scale bars should appear in the images, moreover images seem to be overexposed, the background in fluorescent images must be black to avoid interferences. A representative image of a treated worm should be included.

We agree, that the background of the images is not optimal. The reason for this was probably the long exposure-time. We mentioned the limited validity of this experiment due to possible background interferences in line 720-722. Furthermore, we added representative images for extract treated nematodes as well as scale bars (Figure 5). We believe that - with these caveats brought to the attention of the readers- the analysis is still supporting our conclusion.

14- L 349 image instead of picture

We changed it as suggested (line 395).

15- L 513 improve or extend instead of change

We changed it as suggested (line 553).

16- L 518 please rewrite the sentence: the most efficient survival improvement

With the aid of a native speaker, we improved that sentence (now: line 556-560).

17- L 646 survival instead of survivability

We rewrote this sentence and deleted “survivability” (line 738-744).

18- L 647 compounds instead ingredients

We changed it as suggested (line 744).

19- L664-665 please include the scientific names of the mentioned plants

The scientific names were added as suggested (line 764-765).

20- Appendix A could be included in supplementary material.

As suggested, appendix A is now in the supplementary material (supplementary material 3).

21- L 834 Subtracting the autofluorescence of the controls worms is not the best method to avoid interferences. Splitting the images in RGB and subtracting the red channel to the green channel is a better solution, using lower exposure times works fine too.

We showed in our previous paper that the extract-treatments change the level of red autofluorescence. Nevertheless, also the green autofluorescence was changed due to the extract exposure (unpublished data). Thus, we wanted to avoid measuring differences in autofluorescence instead of differences in the fluorescent DCF. In addition, matched
wild-type controls were also used to correct for autofluorescence in the original publication from Back et al., 2012. Nevertheless, the images were split into RGB by using CellProfiler, which is now mentioned in the respective part in the supplementary material 3.

22- L 856 Indeed the H2DCFDA method is not specific for H2O2, however there are selective probes to quantify superoxide such as hydroethidine

Thanks for this notion. We mentioned this method in the respective paragraph (supplementary material 3).

23- Figure A1 scale bars should appear in the images. A representative image of a treated worm should be included.

We added a scale bar to all nematode images and added representative images of treated worms as well (Fig. 5 and supplementary material 3).

Reviewer 2 Report

In this work, Sayed et al., investigate the effects of two extracts commonly found in TCM blends, which traditionally have been used for their anti-aging effects. They utilize the model organism C. elegans which is a well established model for studying the effects of various drugs on healthspan. They performed fractionation on plant extracts and find that an extract from C. Chinensis provides modest increases in physiological measurements. 

Comments:

- Why were these assays performed at 22C? Most worm aging experiments are performed at 20C. 

- How did the authors decide on the plant extract concentrations for treatment? Would have appreciated a dose response measurement as well as a toxicity assay - at higher concentrations, are these extracts toxic?

- For the survival assay under stress conditions, why did the authors chose 60mM paraquat? Please provide a proper reference for this concentration, citation #30 does not use this concentration. It is useful to use a standardized concentration such that other groups can compare their results. 

ROS measurements - The authors used a 'whole worm' measurement using HyPer. Please cite the original source (Back et al 2012, not the review article). The authors show images for the transgenic animals treated with DMSO or 10mM H2O2. In Back et al (Figure 1 B), they show images where treatment of 10mM H2O2 dramatically affects the sensor. In the author's Figure 5, H2O2 does not seem to do much to the sensor. Please show images for the extract treated worms (this is critical). It would also be useful for the authors to show 'stacks' of worms (ie. multiple worms together in the field of view) to show variability or lack there of.  A scale bar would also be helpful in all images with worms in it. 

- Neuroprotective effects. Since the authors found that C. chinesis improved the mechanosensation of older animals, it would be nice if the authors imaged these neurons (many worm strains are available that label the touch receptor neurons). Aged worms do exhibit morphological changes of these neurons, so it would be interesting to see if this extract can prevent them. Alternatively, since the authors mention synaptic transmission, it would be nice to also image synaptic vesicle markers in these neurons (ie. RAB-3). I am also curious to see if strong antioxidants like NAC can also provide neuro-protective abilities. Please also include an antioxidant as a "negative" control. 

Chemotaxis assays. The authors did not see any change in chemotactic abilities of the worms after treatment - perhaps it's because they only used one concentration? Again, a dose response curve would have been useful here. 

Re: H2DCFDA assay and intracellular ROS. The authors comment that this assay cannot be used as an accurate method to measure intracellular H2O2 since they observed that treatment with their compounds led to an increase in ROS.

In any case, I do not think that the 'increase' in ROS as measured by the DCFDA assay can just be ignored as a negative result. As the authors mention, there is a growing body of literature that suggests that there are protective effects elicited by low doses of increased ROS. A 'whole animal' measurement of fluorescence is not ideal, and I suggest that the authors make better use of the HyPeR transgenes. Since they used microscopy, I think it would be useful if the authors used their Axiolab microscope to image intracellular ROS in certain cells in detail. For example, do the mechanosensory neurons experience less ROS upon treatment with the extract? 

For good measure, please provide all raw data with statistical analyses in your supplemental file. 

Author Response

Dear Ms. Ding, dear Reviewers,

We thank you very much for the constructive suggestions! We tried our best to fulfill all requests. Please find our comments targeting all mentioned issues below:

Reviewer 2 Comments:

- Why were these assays performed at 22C? Most worm aging experiments are performed at 20C. 

The optimal temperature for C. elegans assays ranges between 15 °C and 25 °C. Since the temperature affects (amongst others) the developmental speed, it is important to keep the temperature stable in and between all experiments. Thus, in order to enable a direct comparison with our previous C. cuscuta and E. ulmoides study (Sayed et al., 2021), we kept all maintenance conditions. We included a short note that the maintenance conditions are equal to our previous study (line 159-160). Initially, 22 °C was selected due to pragmatic reasons, because at this temperature the time for the development from the L1 to the L4 stage was optimal for the planning of our experiments.

- How did the authors decide on the plant extract concentrations for treatment? Would have appreciated a dose response measurement as well as a toxicity assay - at higher concentrations, are these extracts toxic?

We did not test the extracts regarding their toxicity, but we assume that they will be toxic in higher concentrations (like most treatments). Initially, we were searching for extracts with health promoting abilities in low concentrations. We hoped that the later use in humans will then also work with low concentrations to reduce side effects and treatment costs. The selected concentration was inspired by several other health- or lifespan promoting extracts, which were successfully used in the range of 25-50 µg/ml in the nematode, such as the Alaskan chaga and cranberry extract (Scerbak et al., 2016), Anacardium occidentale extract (Duangjan et al., 2019), and Korean mistletoe (Lee et al., 2014). This information is now also given in line 170-173.

- For the survival assay under stress conditions, why did the authors chose 60mM paraquat? Please provide a proper reference for this concentration, ci8tation #30 does not use this concentration. It is useful to use a standardized concentration such that other groups can 8compare their results. 

There is not really a standard concentration for using paraquat as an oxidative substance in C. elegans assays. The concentration of paraquat in C. elegans stress assays ranges between 5 mM and 100 mM (For examples, see https://doi.org/10.1155/2019/6840540, https://doi.org/10.1111/j.1474-9726.2006.00192.x,  https://doi.org/10.1042/bj2920605,  https://doi.org/10.1155/2019/7621043, https://doi.org/10.1016/S0891-5849(03)00102-3, and  https://doi.org/10.1371/journal.pone.0008758). Also 60 mM paraquat was used in several published studies, such as https://doi.org/10.4062/biomolther.2013.073, http://dx.doi.org/10.20307/nps.2016.22.3.201, https://doi.org/10.1007/s12272-013-0183-6, and https://doi.org/10.1002/biof.1695. We exchanged citation 30 with proper references (line 227). We decided to use 60 mM paraquat, because the stress-effect was clearly visible, but it was not too strong so that worms did not die too fast and we had enough time to count dead and alive worms several times.

ROS measurements - The authors used a 'whole worm' measurement using HyPer. Please cite the original source (Back et al 2012, not the review article). The authors show images for the transgenic animals treated with DMSO or 10mM H2O2. In Back et al (Figure 1 B), they show images where treatment of 10mM H2O2 dramatically affects the sensor. In the author's Figure 5, H2O2 does not seem to do much to the sensor. Please show images for the extract treated worms (this is critical). It would also be useful for the authors to show 'stacks' of worms (ie. multiple worms together in the field of view) to show variability or lack there of.  A scale bar would also be helpful in all images with worms in it. 

We exchanged the reference as suggested (line 238), added a scale bar to all worm images and selected additional representative worm images including extract treated worms (see Fig. 5). Unfortunately, stack of worms cannot be shown because each worm was photographed separately. The preparation of these images (including the pre-culture and synchronization as well as the treatment with the extracts for 12 days) would need several weeks which exceeds the revision deadline by far.

The difference between the images shown in Back et al. with our images can be explained by several facts. First, we used 12 days old nematodes for the experiment compared to young adults in the Back study. Second, Back and colleagues used a confocal laser scanning microscope, which was not available for our study. Finally, a different maintenance-temperature and other feeding bacteria in unknown concentration were used.

- Neuroprotective effects. Since the authors found that C. chinesis improved the mechanosensation of older animals, it would be nice if the authors imaged these neurons (many worm strains are available that label the touch receptor neurons). Aged worms do exhibit morphological changes of these neurons, so it would be interesting to see if this extract can prevent them. Alternatively, since the authors mention synaptic transmission, it would be nice to also image synaptic vesicle markers in these neurons (ie. RAB-3). I am also curious to see if strong antioxidants like NAC can also provide neuro-protective abilities. Please also include an antioxidant as a "negative" control. 

We agree that these additional tests would be very helpful to understand the neuroprotective role of the extract in detail. Unfortunately, this is beyond the scope of this manuscript. We included your suggestions in the discussion (line 644-653), to give an interesting outlook for the future work in this field.   

Chemotaxis assays. The authors did not see any change in chemotactic abilities of the worms after treatment - perhaps it's because they only used one concentration? Again, a dose response curve would have been useful here. 

Yes, it is possible that another concentration would lead to a beneficial effect in the chemotaxis assay (now mentioned in line 684-687). However, the selected concentration triggered numerous healthspan improvements, thus, it seems to be a question of the selected assay rather than of the extract concentration.

Re: H2DCFDA assay and intracellular ROS. The authors comment that this assay cannot be used as an accurate method to measure intracellular H2O2 since they observed that treatment with their compounds led to an increase in ROS. In any case, I do not think that the 'increase' in ROS as measured by the DCFDA assay can just be ignored as a negative result. As the authors mention, there is a growing body of literature that suggests that there are protective effects elicited by low doses of increased ROS. A 'whole animal' measurement of fluorescence is not ideal, and I suggest that the authors make better use of the HyPeR transgenes. Since they used microscopy, I think it would be useful if the authors used their Axiolab microscope to image intracellular ROS in certain cells in detail. For example, do the mechanosensory neurons experience less ROS upon treatment with the extract? 

Indeed, the measurement with the HyPer strain could be improved. We now added a discussion about the limitation of the HyPer assay as well as a revaluation of the results in concert with the H2DCFDA results (line 720-735).

For good measure, please provide all raw data with statistical analyses in your supplemental file. 

We added all raw data to the supplementary material 4. 

Round 2

Reviewer 1 Report

The authors have improve the manuscript, so congratulation to them

Line 243, I think that 100X magnification is a mistake, for the 200 um scale bar in the images and the shown images are of  whole worms I think that the magnification is 5-10X, please verify the magnification used, is important for the reproducibility of the protocol

Author Response

We calculated the magnification according to the manual of the Axiolab microscope, which states that the total microscope magnification (100x) is the result of the magnification of the eyepiece (10x) multiplied with the magnification of the objective (10x). However, since the total microscope magnification might be confusing, we now changed this part and only mentioned the used magnification of the objective (10x) (line 233-234).

Reviewer 2 Report

Please check over the two paragraphs that were inserted (lines 720-735), it does not flow well and the tone is very informal. Perhaps discuss why measuring whole animal fluorescence is not ideal and why would you look at mechanosensory neurons for these measurements in the first place (because you saw differences in your touch assays). 

I appreciate the changes to Figure 5 where more representative images are inserted. Stylistically, I think the red scale bars are distracting. Maybe a white bar would be better. Additionally, is it possible to change the images to black and white and threshold the background to become black? The blue haze of the background is extremely distracting. 

Author Response

Please check over the two paragraphs that were inserted (lines 720-735), it does not flow well and the tone is very informal. Perhaps discuss why measuring whole animal fluorescence is not ideal and why would you look at mechanosensory neurons for these measurements in the first place (because you saw differences in your touch assays). 

The paragraph was re-checked by a native speaker in order to make the tone more formal. Furthermore, we added a more detailed description why the measurement of certain cells might be better compared with the whole body fluorescence (line 697-722).

I appreciate the changes to Figure 5 where more representative images are inserted. Stylistically, I think the red scale bars are distracting. Maybe a white bar would be better. Additionally, is it possible to change the images to black and white and threshold the background to become black? The blue haze of the background is extremely distracting. 

As suggested, we exchanged the red scale bars with white scale bars (Figure 5). In addition, we tried to convert the images to black-white images. However, we feel that some information gets lost after the conversion. It might be an important information for some readers that the background in the images is blue, thus, we decided to keep the old version. We hope that this is acceptable.